# MEMORY DETERMINES LEARNING DIRECTION: A THEORY OF GRADIENT-BASED OPTIMIZATION IN STATE SPACE MODELS

## ABSTRACT

State space models (SSMs) have gained attention by showing potential to outperform Transformers. However, previous studies have not sufficiently addressed the mechanisms underlying their high performance owing to a lack of theoretical explanation of SSMs' learning dynamics. In this study, we provide such an explanation and propose an improved training strategy. The memory capacity of SSMs can be evaluated by examining how input time series are stored in their current state. Such an examination reveals a tradeoff between memory accuracy and length, as well as the theoretical equivalence between the structured state space sequence model (S4) and a simplified S4 with diagonal recurrent weights. This theoretical foundation allows us to elucidate the learning dynamics, proving the importance of initial parameters. Our analytical results suggest that successful learning requires the initial memory structure to be the longest possible even if memory accuracy may deteriorate or the gradient lose the teacher information. Experiments on tasks requiring long memory confirmed that extending memory is difficult, emphasizing the importance of initialization. Furthermore, we found that fixing recurrent weights can be more advantageous than adapting them because it achieves comparable or even higher performance with faster convergence. Our results provide a new theoretical foundation for SSMs and potentially offer a novel optimization strategy.

## 1 INTRODUCTION

In recent years, machine learning models centered around linear recurrent neural networks (RNNs) have demonstrated strong performance in tasks involving time series modeling (Dao & Gu, 2024). These models, derived from control theory, are referred to as state space models (SSMs). By revealing their learning mechanisms that drive high performance, many studies have sought to develop faster, more energy-efficient, and more accurate models. However, some key techniques introduced in SSMs still lack a solid theoretical foundation and the factors that determine whether gradient-based optimization methods can successfully learn have not been fully identified.

A seminal example of this trend is the high-order polynomial project operators (HiPPO) (Gu et al., 2020a), a family of RNNs that achieves a memory architecture theoretically optimized for the longest possible memory, based on the Lebesgue measure. Building on this foundation, Gu et al. (2022b) proposed the S4 (structured state space sequence model) as a means to implement HiPPO in a computationally efficient way. These studies showed that the initialization of weights plays a critical role in determining final performance (Gu et al., 2020a; 2022b); linear RNNs should be initialized to maximize memory length, even though doing so may inherently degrade memory accuracy because of a theoretical upper bound (Jaeger, 2001a). Subsequently, to further reduce computational overhead, S4D (a simplified S4 with diagonal state matrices) (Gu et al., 2022a) was implemented by simplifying the recurrent weight of the linear RNN into a diagonal matrix. This technique remains at the core of later developments in SSM research (Dao & Gu, 2024; Gu & Dao, 2023). Nevertheless, a theoretical explanation justifying these techniques is still lacking. The reasons why initialization is crucial, and why such a choice of initialization contributes to performance improvement, have only been partly demonstrated (Gu et al., 2022a; Vorontsov et al., 2017). In particular, it remains theoretically unclear whether SSMs can recover from functional failures in a manner analogous to the revival of dead ReLU

neurons (Lu et al., 2020; Horuz et al., 2025). Previous studies suggest that appropriate initialization is important to avoid the failures at initial stages (Vorontsov et al., 2017; Henaff et al., 2016), but this does not imply that improper initialization necessarily leads to a learning failure. If a mechanism existed that allows backpropagation to overcome the failure through learning, then the initialization would be merely a quantitative concern. Moreover, no proof has demonstrated that S4 and S4D are theoretically equivalent. Prior studies theoretically analyzing the learning dynamics of linear RNNs have also not solved these questions (Smékal et al., 2024; Proca et al., 2024; Schuessler et al., 2020). Addressing these issues could lead to better models and more effective training approaches, which may help mitigate typical machine learning problems, including overfitting (Hinton et al., 2012; Salman & Liu, 2019) and long training steps.

To achieve this objective, we can quantitatively evaluate the information processing characteristics of systems in response to input and compare them across models using information processing capacity (IPC) (Dambre et al., 2012). This indicator statistically measures how a time-evolving information processing system transforms arbitrary inputs and retains the processed results within the state as memory, including both linear and nonlinear transformation. IPC can be used for any information processing system that processes inputs over time, regardless of system's internal structure. When calculated with delayed linear inputs, IPC is called memory function (MF) (Jaeger, 2001a; Guan et al., 2025; Rodan & Tino, 2010; Farkaš et al., 2016; Gonon et al., 2020; Ballarin et al., 2024). Because linear RNNs conduct only linear information processing, we can comprehensively investigate the information processing of linear RNNs using MF, which serves as a fundamental and effective metric for evaluating information processing properties in linear models.

In this study, we clarify the learning mechanism of SSMs in three stages, specifically by focusing on S4 and S4D. First, we reformulate the SSM by retaining only the essential components and demonstrate that MF plays a crucial role in SSMs; this establishes a foundation for analyzing SSMs through the lens of MF. Second, by analytically examining the gradients in an SSM layer, we reveal how the initial structure of the MF influences the learning process; this highlights the essential role of initialization and shows that, for successful learning, the initial MF must prioritize memory length over precision or crucial label information may be lost during backpropagation through a single SSM layer. Combining this theoretical result and the results of MF, we propose that the linear recurrent weight may not require training if SSMs are initialized with weights that satisfy the memory requirements. Finally, we empirically validate our theoretical results by evaluating performance across all tasks in the long range arena (LRA) benchmark (Tay et al., 2020) and demonstrate that our proposed method is effective. The results show faster convergence, mitigation of overfitting, and the potential for further improvements in accuracy.

## 2 BACKGROUND

### 2.1 STRUCTURED STATE SPACE SEQUENCE MODEL

S4 is a deep learning architecture designed to efficiently capture long-range dependencies in sequential data. It builds on continuous-time linear SSMs, particularly those formulated in the HiPPO framework, and translates them into trainable and hardware-efficient architecture suitable for modern sequence modeling tasks. S4 begins with the general form of a continuous-time SSM:

$$\dot{\boldsymbol{x}}(t) = \tilde{\boldsymbol{A}}\boldsymbol{x}(t) + \tilde{\boldsymbol{B}}\boldsymbol{u}(t) \qquad \boldsymbol{y}(t) = \tilde{\boldsymbol{C}}\boldsymbol{x}(t) + \tilde{\boldsymbol{D}}\boldsymbol{u}(t), \tag{1}$$

where $\boldsymbol{x}(t) \in \mathbb{R}^N$ is the state, $\boldsymbol{u}(t) \in \mathbb{R}$ is the input, and $\boldsymbol{y}(t) \in \mathbb{R}$ is the output. The matrices $\tilde{\boldsymbol{A}} \in \mathbb{R}^{N \times N}$, $\tilde{\boldsymbol{B}} \in \mathbb{R}^{N \times 1}$, $\tilde{\boldsymbol{C}} \in \mathbb{R}^{1 \times N}$, and $\tilde{\boldsymbol{D}} \in \mathbb{R}$ define the system dynamics and output behavior. In a prior study, HiPPO matrices were adopted as $\tilde{\boldsymbol{A}}$ and $\tilde{\boldsymbol{B}}$, theoretically realizing the longest memory structure (Gu et al., 2020b). In S4D, "S4Dinv" and "S4Dlin" were introduced by approximating the eigenvalues of the HiPPO matrices (Gu et al., 2022a) (see Appendix B). For practice, the continuous model (Eq. 1) is discretized as follows:

$$\boldsymbol{x}_{k+1} = \boldsymbol{A}\boldsymbol{x}_k + \boldsymbol{B}\boldsymbol{u}_k \qquad \boldsymbol{y}_k = \tilde{\boldsymbol{C}}\boldsymbol{x}_k + \tilde{\boldsymbol{D}}\boldsymbol{u}_k, \tag{2}$$

where $\boldsymbol{x}_k$, $\boldsymbol{u}_k$, and $\boldsymbol{y}_k$ denote the state, input, and output at discrete time $k$, respectively. The detailed discretization procedure of the variables and parameters $\boldsymbol{A}$ and $\boldsymbol{B}$ is described in Appendix B. The S4 architecture extends the basic SSM by enabling both parallelization and deep stacking across multiple layers. In the parallel setting, a sequence of inputs $\{\boldsymbol{u}_k\}_{k=1}^T$ is processed by applying the discretized

state-space recurrence to each element independently using fast convolution methods, yielding an output sequence $\{\boldsymbol{y}_k\}_{k=1}^T$. To construct a deep S4 model, multiple such layers are stacked, with each layer consisting of a structured SSM followed by a pointwise nonlinearity and a linear transformation. Let $\boldsymbol{u}_k^{(\ell)}$ denote the output at time step $k$ in layer $\ell$. Then, the $\ell$th layer of the S4 network is defined by

$$\boldsymbol{x}_{k+1}^{(\ell)} = \boldsymbol{A}^{(\ell)}\boldsymbol{x}_k^{(\ell)} + \boldsymbol{B}^{(\ell)}\boldsymbol{u}_k^{(\ell-1)} \qquad \boldsymbol{u}_k^{(\ell)} = \boldsymbol{W}^{(\ell)}f(\boldsymbol{y}_k^{(\ell)}) + \boldsymbol{b}^{(\ell)}, \tag{3}$$

where $(\cdot)^{(\ell)}$ denotes the value in $\ell$th layer, $f(\cdot)$ is a nonlinear activation function [e.g., GELU or GLU (Hendrycks & Gimpel, 2016; Dauphin et al., 2017)], and $\boldsymbol{W}^{(\ell)}$ and $\boldsymbol{b}^{(\ell)}$ are the weight and bias parameters, respectively, of a linear transformation applied after the state-space computation.

## 2.2 MEMORY CAPACITY

We evaluate the information processing in the discretized SSM using MF (Jaeger, 2001a; Dambre et al., 2012), which represents how well the delayed input series $u_{t-\tau}$ can be reconstructed from the current network state. Any i.i.d. input can be employed when evaluating the MF. Following previous studies, we adopt the uniformly random input $u_t \in [-1, 1]$ in numerical experiments. An emulation is conducted by a linear approximation: $\hat{u}_{t-\tau} = \boldsymbol{w}_{\text{out}}^\top\boldsymbol{x}_t$, where $\tau$ is the delay from the current time. The readout weight vector $\boldsymbol{w}_{\text{out}}$ is determined by minimizing a loss function of the mean squared error (MSE): $L_{\text{MSE}} = \frac{1}{T}\sum_{t=1}^T(\hat{u}_{t-\tau} - u_{t-\tau})^2$, where $T$ is the sampled time length. Accordingly, the MF is defined as follows:

$$M[u_{t-\tau}] = 1 - \frac{\min_{\boldsymbol{w}_{\text{out}}}\langle(\hat{u}_{t-\tau} - u_{t-\tau})^2\rangle}{\langle u_{t-\tau}^2\rangle} \ (\leq 1), \tag{4}$$

where $\langle\cdot\rangle$ denotes the time average. The upper bound 1 is satisfied when the system has fully memorized the target, which is the delayed input series. The sum of the MF with respect to all $\tau$ represents the memory capacity (MC) of the SSM:

$$M_{\text{sum},u} = \sum_{\tau=0}^\infty M[u_{t-\tau}] \ (\leq N). \tag{5}$$

The upper bound of $M_{\text{sum},u}$ is determined by the number of linearly dependent time series in the state, which is called the rank and is ideally $N$. This implies that there is a tradeoff between the accuracy and length of memory. By calculating Eq. (4), Dambre et al. (2012) introduced another expression for the MF: $M[u_{t-\tau}] = \boldsymbol{U}_\tau^\top\boldsymbol{X}(\boldsymbol{X}^\top\boldsymbol{X})^{-1}\boldsymbol{X}^\top\boldsymbol{U}_\tau/(\boldsymbol{U}_\tau^\top\boldsymbol{U}_\tau)$, where $\boldsymbol{X} \in \mathbb{R}^{T \times N}$ is a matrix whose columns represents the state time series, and $\boldsymbol{U}_\tau \in \mathbb{R}^T$ is the delayed input series. The MF and MC of linear RNNs were analyzed in a previous study (Guan et al., 2025):

$$M[u_{t-\tau}] = \boldsymbol{V}_\tau^\top(\boldsymbol{V}^\top\boldsymbol{V})^{-1}\boldsymbol{V}_\tau \qquad M_{\text{sum},u} = \text{tr}\left[\boldsymbol{V}(\boldsymbol{V}^\top\boldsymbol{V})^{-1}\boldsymbol{V}^\top\right], \tag{6}$$

where $\boldsymbol{V} = \begin{pmatrix}\boldsymbol{V}_{T-1} & \cdots & \boldsymbol{V}_1 & \boldsymbol{V}_0\end{pmatrix}^\top$ is a Vandermonde matrix, $\boldsymbol{V}_\tau = \begin{pmatrix}\lambda_1^\tau & \lambda_2^\tau & \cdots & \lambda_N^\tau\end{pmatrix}^\top$, and $\{\lambda_i\}_{i=1}^N$ are the eigenvalues of matrix $\boldsymbol{A}$ in Eq. 2. The maximum absolute eigenvalue is called the spectral radius, which is a crucial parameter for the dynamics of linear RNNs. If this value is greater than 1, the states will diverge as the time step progresses.

## 2.3 GRADIENT BASED LEARNING

One of the most widely used parameter update strategies in machine learning models is based on the principle of gradient descent. Variants such as stochastic gradient descent (SGD) (Robbins & Monro, 1951; Bottou et al., 1991) and Adam (Kingma & Ba, 2014) exist. In all of these methods, errors propagate from the final layer backward, and the gradients computed at each layer determine the direction of parameter updates. Here, the error at the final layer is defined as $L_o(\hat{\boldsymbol{y}}_o, \boldsymbol{y}_o)$, where $\hat{\boldsymbol{y}}_o \in \mathbb{R}^{H_o}$ is the model output, $\boldsymbol{y}_o \in \mathbb{R}^{H_o}$ is the target label, and $H_o$ is the dimensionality of the desired output. In gradient-based learning, to evaluate the error of the previous layer and determine the parameter update direction, the gradient $\nabla_{\boldsymbol{\theta}_o}L_o(\hat{\boldsymbol{y}}_o, \boldsymbol{y}_o)$ of the final layer is calculated, where $\boldsymbol{\theta}_o$ denotes the parameters of the output layer. Subsequently, the gradient for the layer immediately preceding the output layer is computed, a process that is applied recursively across layers and forms the basis for parameter updates. Accordingly, the optimal output of $l$th layer is determined by the

gradient passed from the $(l+1)$th layer, suggesting that we can define the loss function $L_l(\hat{\boldsymbol{y}}_l, \boldsymbol{y}_l)$ for the $l$th layer in a manner similar to the output layer, where $\hat{\boldsymbol{y}}_l$ and $\boldsymbol{y}_l$ represent the layer output and the ideal outputs of that layer.

In this study, we assume that the loss function $L_l$ of the SSM layer can be represented by the MSE. Accordingly, the gradient of the SSM is described by

$$\nabla_{\boldsymbol{\theta}_l} L_l(\hat{\boldsymbol{y}}_l, \boldsymbol{y}_l) = \nabla_{\boldsymbol{\theta}_l} \frac{1}{H_l} ||\hat{\boldsymbol{y}}_l - \boldsymbol{y}_l||^2, \tag{7}$$

where $\boldsymbol{\theta}_l$ denotes the parameters of $l$th layer and $H_l$ is the dimensionality of the layer output.

## 3 RESULTS

### 3.1 MF OF SSM

In this section, we show why MF is an effective indicator for SSMs in two steps. First, we analytically simplify the SSM based on the theoretical results of MF obtained in previous studies. Second, we numerically demonstrate the MF of SSM architectures proposed by prior research and explain their memory profiles in terms of MF.

**S4D is theoretically equivalent to S4**   According to a previous study (Dambre et al., 2012), all information processing of linear SSMs can be comprehensively measured by MF. The state $\boldsymbol{x}_t$ of linear RNNs can be described by $\boldsymbol{x}_t = \sum_{\tau=0}^{\infty} \boldsymbol{\alpha}_\tau u_{t-\tau}$, where $\{\boldsymbol{\alpha}_\tau\}_{\tau=0}^{\infty}$ is a certain series of fixed constant vectors as in Eq. 20. This indicates that the output $y_t$ from the readout is represented by $y_t = \sum_{\tau=0}^{\infty} \alpha_\tau u_{t-\tau}$ using a constant sequence $\{\alpha_\tau\}_{\tau=0}^{\infty}$, suggesting that linear RNNs perform information processing solely by computing a linear combination of past inputs. Since the MF $M[u_{t-\tau}]$ is proven to be analytically equivalent to the norm of $\boldsymbol{\alpha}_\tau$ (Kubota et al., 2021), it can fully capture the information processing characteristics of linear SSMs. Combining this result with the analytical solution obtained from Guan et al. (2025) (Eq. 6) reveals that processed inputs held in the SSM are determined only by the eigenvalues of matrix $\boldsymbol{A}$ (Eq. 2). This proves that both the input weight $\boldsymbol{B}$ and the eigenvectors of $\boldsymbol{A}$ can be freely specified, demonstrating that S4 is theoretically equivalent to S4D because they share the same eigenvalues. Accordingly, we redefine the recurrent equation of the discrete SSM:

$$\boldsymbol{x}_{k+1} = \boldsymbol{\Lambda}\boldsymbol{x}_k + \boldsymbol{B}u_k, \tag{8}$$

where $\boldsymbol{\Lambda} = \text{diag}\begin{pmatrix} \lambda_1 & \lambda_2 & \cdots & \lambda_N \end{pmatrix}$. Unless otherwise specified, we use an input weight $\boldsymbol{B} = B\boldsymbol{1}$, where $\boldsymbol{1} \in \mathbb{R}^{N \times 1}$ is the vector whose elements are all 1 and $B$ is an arbitrary value.

**MF of HiPPO eigenvalues**   To demonstrate the effectiveness of MF, we calculated it for five typical eigenvalue realizations and analyzed the corresponding memory structures. Two groups of eigenvalues for the SSMs are employed. The first group consisted of those proposed in previous studies as realizing the longest memory structure—namely, "S4Dinv" and "S4Dlin." The second group consisted of other representative eigenvalues: "random", "lin", and "step." "random" refers to eigenvalues of matrix $\boldsymbol{A}$ randomly generated from a uniform distribution. "lin" indicates real eigenvalues that are completely linearly distributed. "step" indicates that all eigenvalues are complex conjugate pairs whose magnitudes are arranged in a linear distribution. We evaluated the MFs based on the actual setting used for training and evaluation of the model. In principle, one SSM layer consists of multiple distinct SSMs. To evaluate the information processing of the entire SSM, we defined the supremum MF of the $l$th layer:

$$M[u_{t-\tau}]^{(\ell)} = \sup M[u_{t-\tau}]_i^{(\ell)} = \max_i M[u_{t-\tau}]_i^{(\ell)}, \tag{9}$$

where $i$ is the index identifying a certain SSM. Figure 1 shows the eigenvalues and the MF of one layer (Eq. 9), and it illustrates that "S4Dinv" and "S4Dlin" exhibited longer and stronger memory than other eigenvalue realizations, suggesting they possess richer information processing. Clearly, the sums of MFs differ. This can be explained by the fact that the MF sum numerically does not reach the upper bound, number of nodes $N$, and the system ranks are different between systems defined by the eigenvalues. Therefore, the longer MFs are attributed to the characteristics of eigenvalues; "S4Dinv" and "S4Dlin" inherently generate higher system ranks. From the perspective of the MF, this eigenvalue group can be considered superior to the other group. We provide the motivation for choosing these configurations and the detailed procedure of analysis in Appendices B and C.

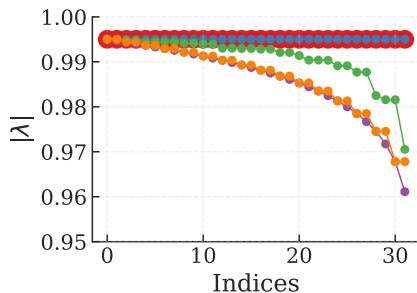 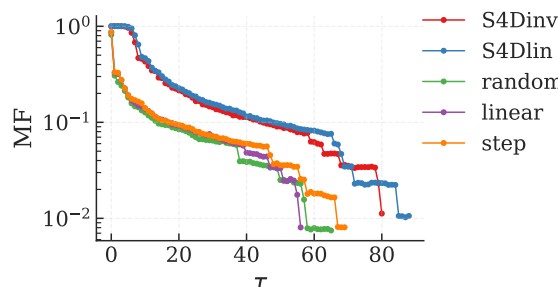

Figure 1: The MF of five eigenvalue realizations. The left (right) panel shows the absolute eigenvalues (MF). The horizontal axis is the index of eigenvalues (delay $\tau$ of input). The system size $N$ is 32, and the results were obtained by calculating Eq. 4, where $T = 1024$ was typically used in previous studies' models.

### 3.2 IMPORTANCE OF INITIAL PARAMETERS FOR SUCCESSFUL LEARNING

Next, we investigate the learning process by focusing on the parameter update method. To simplify the problem, we introduce two assumptions: (i) the input $u_t$ is randomly sampled from a uniform distribution, consistent with the setting used for the MF; (ii) we consider the case where matrix $\boldsymbol{A}$ is fixed at its initial value. Assuming that the SSM has an ideal output series $\boldsymbol{y} = \{y_i\}_{i=1}^T$, the loss function of single SSM layer is described by $L_{\text{MSE}} = \frac{1}{T}\sum_{i=1}^{T}(f(\boldsymbol{\theta}^\top \boldsymbol{x}_i) - y_i)^2$, where $T$ is the output dimension of SSM, $\boldsymbol{\theta} = \boldsymbol{W}^{(\ell)}$ is the parameter of the linear readout layer, $f(\cdot)$ is the piecewise activation function applied after the linear readout, and $\boldsymbol{x}_i$ is the states used to produce each $y_i$. The gradient $G(\boldsymbol{y})$ can be computed as

$$G(\boldsymbol{y}) = \nabla_\theta L_{\text{MSE}} = \frac{2}{T}\boldsymbol{X}^\top\left[(f(\boldsymbol{X}\boldsymbol{\theta}) - \boldsymbol{y}) \odot f'(\boldsymbol{X}\boldsymbol{\theta})\right]. \tag{10}$$

To analyze what affects the parameter update process intuitively, we focus on the SSM before the nonlinear readout is applied. In the case where $f$ is linear, the gradient is described as follows:

$$G(\boldsymbol{y}) = \frac{2}{T}\boldsymbol{X}^\top(\boldsymbol{X}\boldsymbol{\theta} - \boldsymbol{y}). \tag{11}$$

Following the approach of the MF, we consider a task to predict a delayed input series $\boldsymbol{y} = \boldsymbol{U}_\tau \in \mathbb{R}^T$. The gradient in this task is computed as $G(\boldsymbol{U}_\tau) = 2\langle u^2\rangle B\boldsymbol{V}^\top(B\boldsymbol{V}\boldsymbol{\theta} - \boldsymbol{V}(\boldsymbol{V}^\top\boldsymbol{V})^{-1}\boldsymbol{V}_\tau)$. Using the fact that linear SSM performs only linear information processing, the linear combination of past input series $\boldsymbol{y} = \sum_{k=0}^{T-1}\alpha_k\boldsymbol{U}_k$ fully describes the possible outputs of the SSM, where $\{\alpha_\tau\}_{\tau=0}^{T-1}$ is a series characterizing $\boldsymbol{y}$. Accordingly, the gradient $G(\boldsymbol{y})$ calculated with this output $\boldsymbol{y}$ describes all possible gradients of the SSM layer (Lemma D.2). The gradient is computed as

$$G(\boldsymbol{y}) = 2\langle u^2\rangle B\boldsymbol{V}^\top(B\boldsymbol{V}\boldsymbol{\theta} - \boldsymbol{V}(\boldsymbol{V}^\top\boldsymbol{V})^{-1}\boldsymbol{V}^\top\boldsymbol{\alpha}). \tag{12}$$

The derivation of these formulas is shown in Appendix D. According to Eq. 12, two important insights are revealed. First, the state series $\boldsymbol{X}$ (Eq. 11) is converted to the matrix $B\boldsymbol{V}$, which suggests that $\boldsymbol{V}$ essentially represents the state series during parameter updates. Second, the matrix $\boldsymbol{V}(\boldsymbol{V}^\top\boldsymbol{V})^{-1}\boldsymbol{V}^\top$, which is the normalized version of $\boldsymbol{V}^\top$, acts as a weighting function for $\boldsymbol{\alpha}$. The $\tau$th element of the vector $\boldsymbol{\alpha}$ is multiplied with the $\tau$th column of $\boldsymbol{V}^\top$. The $\tau$th column of $\boldsymbol{V}^\top$, denoted as the vector $\boldsymbol{V}_\tau$, represents the decay of input information from $\tau$ steps ago. From this, it is theoretically shown that the decay of past input information in SSMs determines how much information from $\tau$ steps ago affects the parameter update. Because the decay of input information stored in the state time series can be evaluated by the MF, and because $\boldsymbol{V}$ represents $\boldsymbol{X}$ in the gradient computation, the decay represented by $\boldsymbol{V}$ can be evaluated through the MF. In fact, the diagonal components of the matrix $\boldsymbol{V}(\boldsymbol{V}^\top\boldsymbol{V})^{-1}\boldsymbol{V}^\top$ are the MF. Therefore, if the teacher time series $\boldsymbol{y}$ contains past input information whose memory is evaluated as 0 through MF, such information is not reflected in the learning process.

**Memory requirement that must be realized at initialization** Consider an extreme case where $\boldsymbol{y}$ is solely composed of inputs from the distant past such that the MF indicates approximately 0. In this case, because

$$\boldsymbol{V}(\boldsymbol{V}^\top\boldsymbol{V})^{-1}\boldsymbol{V}^\top\boldsymbol{\alpha} \approx \boldsymbol{0}, \tag{13}$$

the parameter update is independent of the desired output, the learning process ideally does not proceed. As there is no explicit mechanism to update the MF through the learning (Theorem D.4), the initialization of eigenvalues is crucial. To ensure an update reflecting the teacher information, we must initialize the weight with an MF that meets the following condition: the values of MF remain sufficiently larger than 0 in the distant past such that all $\alpha_k$ are not 0. The result reveals a crucial requirement for SSMs to solve universal tasks. To learn tasks that rely on information from arbitrarily distant past inputs, the SSM must be initialized with an infinitely long memory (Lemma D.5). According to the tradeoff between the accuracy and length of memory (Eq. 5), we must satisfy this condition by abandoning the accuracy of memory to some extent. Our result suggests that this condition may be given higher priority over the memory accuracy because gradient updates are still executed using teacher signals, even when the memory is very weak (Theorem D.6). However, if accuracy is prioritized, teacher signals from the distant past are likely to be completely discarded, and effective updates may not progress at all. The meaning of the product $V(V^\top V)^{-1}V^\top\alpha$ is further analyzed in Appendix D.3.

**SSM with nonlinear output layer**  The analysis restricted to the linear readout can fully account for the nonlinear readout case because the input of readout layer is entirely dependent on the output of SSM. However, to illustrate how nonlinearity affects the output of the full SSM layer, we provide the formula for nonlinear readout here (Theorem D.1). In the case where $f$ is nonlinear, the gradient (Eq. 10) is described by

$$G(\boldsymbol{y}) \;=\; \frac{2}{B}\boldsymbol{X}^\top\left[f(\boldsymbol{X\theta})\odot f'(\boldsymbol{X\theta}) - \hat{\boldsymbol{U}}^\top\boldsymbol{V}(\boldsymbol{V}^\top\boldsymbol{V})^{-1}\boldsymbol{V}^\top\bar{\boldsymbol{\alpha}}\right], \tag{14}$$

where $\hat{\boldsymbol{U}} = (\boldsymbol{U}_{T-1}\;\;\cdots\;\;\boldsymbol{U}_0)$. The vector $\bar{\boldsymbol{\alpha}}$ is the series characterizing the label $\bar{\boldsymbol{y}} = \boldsymbol{y}\odot f'(\boldsymbol{X\theta})$, modified by the derivative of the activation function, assuming the input $u_t$ can be considered the source of $\boldsymbol{y}$. We can confirm that the update direction is determined by the vector $\boldsymbol{V}(\boldsymbol{V}^\top\boldsymbol{V})^{-1}\boldsymbol{V}^\top\bar{\boldsymbol{\alpha}}$. Therefore, we can elucidate the update direction by the product of MF and the modified label series similarly. The same result applies in the nonlinear case.

**Training eigenvalues**  The risk of learning is implied through our analysis. Here we remove assumption (ii) and allow the SSM layer to learn the eigenvalues of the internal weight matrix. In this case, the MF changes during the training process, and the memory within certain time regions may be lost because the sum of the MF is theoretically constant (Eq. 5). To increase the MF within a specific time region, it is theoretically necessary to reduce the MF in other regions. Even numerically, we cannot ensure that the lost memory will not be required again during training. For example, when we use the structured eigenvalues [e.g., "S4Dinv" and "S4Dlin" (Gu et al., 2022a)], their theoretically longest memory structure can be destroyed. On the other hand, the advantages of learning eigenvalues can also be explained using MF in terms of both shortening and extending memory. Shortening the MF is beneficial in tasks where only short memory is required (Conjecture D.7). It is preferable to concentrate on essential memory rather than sustain low-precision memory over long delays, because the sum of the MF is theoretically fixed. The benefits of extending the MF have already been discussed above. As already clarified, gradient-based learning does not possess any explicit mechanism to deliberately extend the MF in regions where no memory is present. Nevertheless, as eigenvalues are adapted to alter the organization of existing memory, an unintended extension of the MF (Definition D.8) may also occur (Conjecture D.9). For example, when the MF at $\tau = 80$ is enhanced during learning with eigenvalues of "S4Dinv" in Figure 1, the MF at $\tau = 100$, which is initially 0, may also emerge, though this is not guaranteed by the existing algorithm. To determine which learning strategy is more effective, numerical experiments are conducted in the next section.

## 3.3 SSM with fixed eigenvalues

To address the question raised in the previous section, we propose the use of an SSM with fixed eigenvalues and only learning the output weight of each SSM. We refer to this setup as the reservoir computing (RC) setting, following the RC approach in which internal layer weights remain fixed and only output weights are learned  (Jaeger, 2001b; Maass et al., 2002; Lukoševičius & Jaeger, 2009; Nakajima & Fischer, 2021). This approach not only preserves the memory structure but also addresses common issues encountered in machine learning, such as overfitting owing to a large number of trainable parameters, potentially impairing generalization performance. To demonstrate the effectiveness of this learning method and to investigate how initializing with the longest memory structure affects learning, we evaluate all five eigenvalue initializations for the SSM. As benchmark

tasks, we employed the LRA benchmark, which is widely used in prior work (Gu et al., 2022b;a) and requires long-term memory to solve. For all tasks and under each eigenvalue realization, we computed task accuracies in two settings: (1) a setting in which eigenvalues are allowed to be learned and (2) the RC setting. To examine the robustness of each model, we trained them with six different random seeds. Table 1 presents the maximum and minimum accuracies obtained under each condition. Except for the random seed, all model hyperparameters were kept identical to those used in previous work (Gu et al., 2022b). Detailed values are provided in Appendix E.

Table 1: Test accuracy of S4D on LRA tasks with different eigenvalue initializations in the SSM. Each task was executed six times with different random seeds to investigate whether the models were robust. The minimum and maximum accuracies across runs are reported. For reference, we include the S4 results reported in the official repository (Gu et al., 2022b). Three settings are presented: "S4", "learnable eigenvalues", and "RC." Accuracies are underlined when a given eigenvalue realization achieves the highest accuracy across each setting. In addition, when "RC" surpasses "learnable eigenvalues," the corresponding accuracy is bolded.

| | Listops | | Text | | Retrieval | | Image | | Pathfinder | | PathX | |
|---|---|---|---|---|---|---|---|---|---|---|---|---|
| Eigenvalues | max acc | min acc | max acc | min acc | max acc | min acc | max acc | min acc | max acc | min acc | max acc | min acc |
| S4 | 0.595 | | 0.865 | | 0.910 | | 0.885 | | 0.940 | | 0.960 | |
| S4Dinv | 0.594 | 0.589 | 0.853 | 0.839 | 0.863 | 0.760 | 0.864 | 0.859 | 0.910 | 0.877 | 0.922 | 0.903 |
| S4Dlin | 0.598 | 0.587 | 0.848 | 0.830 | 0.843 | 0.750 | 0.868 | 0.860 | 0.896 | 0.865 | 0.926 | 0.912 |
| random | 0.552 | 0.476 | 0.857 | 0.835 | 0.903 | 0.899 | 0.820 | 0.815 | 0.870 | 0.846 | 0.877 | 0.841 |
| linear | 0.543 | 0.488 | 0.839 | 0.811 | 0.899 | 0.894 | 0.782 | 0.768 | 0.797 | 0.754 | 0.820 | 0.503 |
| step | 0.605 | 0.575 | 0.862 | 0.852 | 0.883 | 0.841 | 0.789 | 0.779 | 0.858 | 0.824 | 0.840 | 0.824 |
| RC S4Dinv | 0.585 | 0.570 | **0.857** | **0.846** | **0.906** | **0.901** | **0.873** | **0.865** | 0.899 | **0.889** | 0.932 | 0.923 |
| RC S4Dlin | 0.593 | 0.577 | **0.850** | **0.843** | **0.903** | **0.898** | **0.877** | **0.871** | 0.892 | **0.872** | 0.933 | 0.922 |
| RC random | 0.535 | **0.477** | 0.845 | 0.813 | 0.901 | 0.896 | 0.733 | 0.698 | 0.824 | 0.777 | 0.820 | 0.773 |
| RC linear | 0.530 | 0.423 | 0.815 | 0.801 | 0.894 | 0.889 | 0.686 | 0.634 | 0.769 | 0.506 | 0.669 | **0.504** |
| RC step | 0.556 | 0.500 | 0.845 | 0.833 | **0.903** | **0.897** | 0.745 | 0.740 | 0.832 | 0.793 | 0.813 | 0.504 |

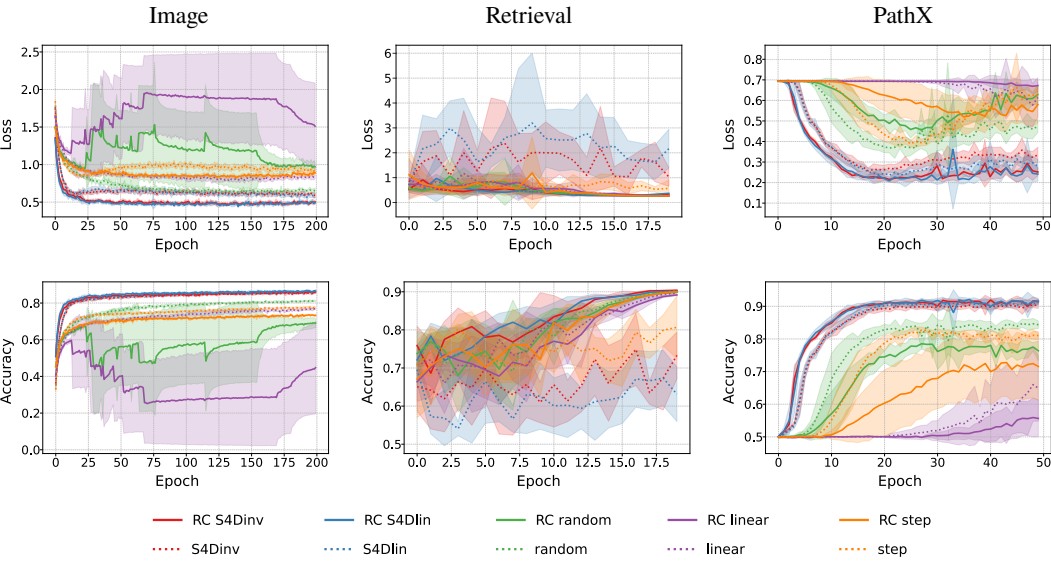

Figure 2: Test loss and accuracy of three tasks. The top (bottom) row indicates test loss (accuracy), and each column corresponds to a different task. The horizontal axes represent epochs. Lines represent averages across seeds, with shaded regions indicating standard deviations. Solid lines represent models in the RC setting, while dotted lines indicate models with trainable eigenvalues. Colors indicate different eigenvalue realizations.

Our experiments show that the RC setting is effective across all LRA tasks when the eigenvalues are initialized with structured memory. In five tasks, the RC setting with structured eigenvalues yielded higher accuracy than the setting with learnable eigenvalues and achieved accuracies comparable to those of the S4 model. In contrast, the model with representative eigenvalues showed poorer performance under the RC setting, suggesting the importance of the initialization with structured

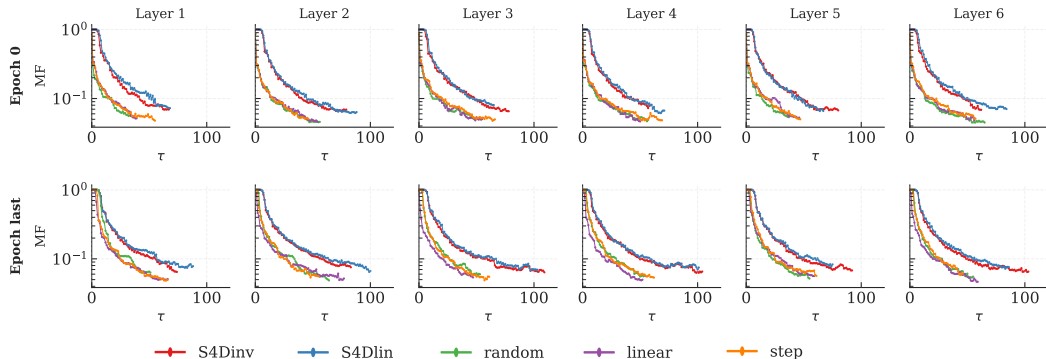

Figure 3: The supremum MF of each layer before and after training. The top (bottom) panels show MFs before (after) training. From left to right, each column represents one layer. The horizontal (vertical) axis shows the input delay $\tau$ (MF). The five colors indicate the eigenvalue realizations. The state size $N$ is 32, and the results are obtained by calculating Eq. 4 with $T = 1024$. As an example, we show the results for the Pathfinder task.

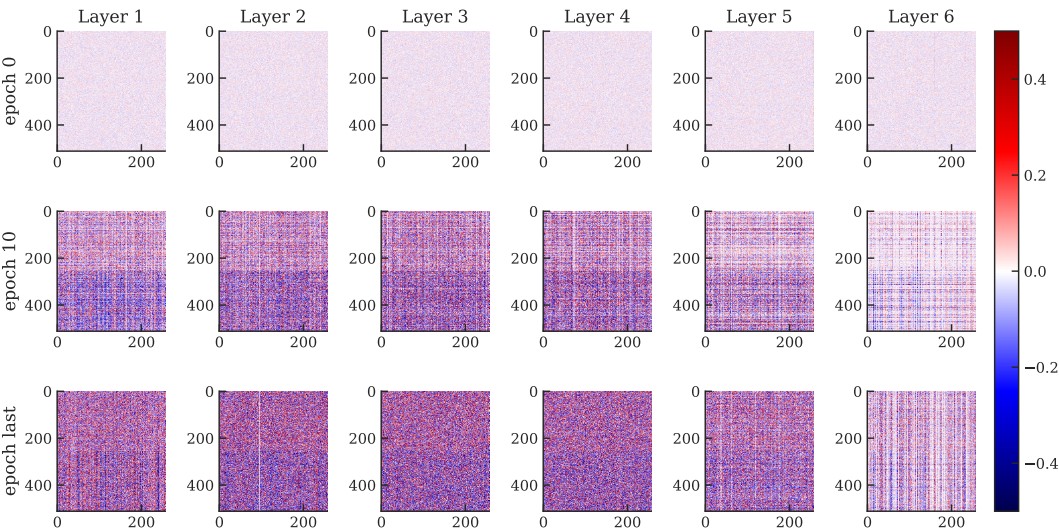

Figure 4: The linear readout weights $\boldsymbol{W}^{(\ell)}$ and $\boldsymbol{b}^{(\ell)}$ at each layer and epoch. From top to bottom, the rows correspond to epochs 0, 10, and the final epoch, respectively. From left to right, the columns show the weights of layers 1 through 6. Following Figure 3, we present an example from the Pathfinder task.

eigenvalue realizations. To examine the effects of the RC setting, we monitored how the loss and accuracy evolved over time. Using structured eigenvalues, the test loss for Image, Retrieval, and PathX converged to lower values more rapidly in the RC setting, and the test accuracy for Retrieval and PathX also increased notably faster (Figure 2). In addition, for the Text, Retrieval, and Image tasks, the RC setting yielded test accuracy comparable—to or even exceeding—those obtained when eigenvalues were learned. Moreover, the training accuracy shows a relatively modest enhancement with RC setting, indicating better mitigation of overfitting (see Appendix E.3). For the remaining tasks, these trends were less pronounced, yet no significant contradictory outcomes were observed. For example, the degradation in ListOps performance can be explained by a modest memory improvement obtained through eigenvalue learning, which outweighs the benefit of fixed eigenvalues. These results suggest that fixing eigenvalues during learning contributes to more efficient training.

To observe whether eigenvalue updates are necessary in detail, we examined how the MF and readout weight of SSMs change before and after training when learning is allowed. As shown in Figure 3, even after training, the MFs of the "random", "linear", and "step" eigenvalue configurations did

not surpass those of structured eigenvalues before training. This implies the difficulty of training eigenvalues and highlights the importance of initialization. Focusing on each eigenvalue realization, training was able to extend the MF, suggesting the benefits of trainable eigenvalues. However, as seen in the accuracies (Table 1), the RC setting yielded better performance, suggesting that learning eigenvalues may offer fewer benefits than the RC setting if the initial eigenvalues already achieve a relatively long MF. The reason why the extended MFs did not contribute to performance can be attributed to the factors mentioned above—namely, the suppression of overfitting and stabilized loss at lower values. In Figure 4, the dynamics of the weights in the linear output layer are shown when eigenvalue training is allowed. They change significantly over the course of training, indicating that the readout learning is important even when the eigenvalues are trained. Combined with the high scores of the RC setting, this observation suggests that learning in SSMs may primarily progress through parameters other than eigenvalues.

## 4 DISCUSSION AND CONCLUSION

In this study, we theoretically demonstrated the importance of initialization and the memory requirement for successful learning, which were not addressed in previous studies attempting to elucidate the learning mechanisms of SSMs. In addition, those studies imposed constraints on $A$ without sufficient theoretical justification. For example, the results of the original S4D study (Gu et al., 2022a) apply only to $A$ in HiPPO framework, $A$ was taken as random matrices (Schuessler et al., 2020), and the eigenvectors of $A$ were either restricted or simply ignored (Smékal et al., 2024; Proca et al., 2024). Restricting eigenvectors limits applicability to specific models, and ignoring them confines the results to S4D, whose results cannot be applied to S4. In contrast, this study imposes no restrictions on $A$, thereby extending the applicability of our results to a wider range of models. We theoretically justify that the eigenvectors of $A$ can be disregarded, allowing us to focus solely on the eigenvalues. The broader applicability of our results is discussed in Appendix F.4. Furthermore, because our results rely only on gradients, they are not limited to any specific training method and can be applied to any gradient-based optimization approach.

Utilizing the theoretical base, we focus on identifying the factors that determine whether gradient-based optimization methods can successfully learn. To this end, we considered how inputs affect the gradient. As a result, it is revealed that the MF, describing the degree of decay over a delay of $\tau$, weights the teacher information corresponding to the $\tau$-delayed input, Therefore, MF has the role of selecting the teacher components, suggesting the mechanism of losing teacher information. This naturally leads to the question of whether BP can recover from such a failure. Our results show that BP can solve this problem only in limited cases. Using the MF, we proved that if a memory about a certain input exist initially, the error information regarding the input may be improved by increasing the memory accuracy. But if the memory of the input does not exist initially, the error information cannot be explicitly improved at all. This arises from the fact that BP has no explicit mechanism to extend the memory structure. Because this is the problem of BP, we argue that our study could reveal a part of learning mechanism.

Related research topics include the vanishing and exploding gradient problems (Zucchet & Orvieto, 2024; Park et al., 2023; Sokół et al., 2019). Previous studies suggest that appropriate initialization is important to avoid gradient vanishing (Vorontsov et al., 2017), but this does not imply that improper initialization necessarily leads to learning failure. If a mechanism existed that allows backpropagation to overcome vanishing gradients and prevents them from occurring through learning, then gradient vanishing at the initial stage would not be problematic. In prior studies on ReLU neurons, some methods have been proposed to reduce the number of dead neurons through initialization (Lu et al., 2020) and to revive neurons that have already died (Horuz et al., 2025). Such issues have not been discussed in the context of SSMs; however, they are important to explain why the SSM succeeds in learning. Our results show that SSMs do not have mechanism to overcome the loss of information when using any gradient-based optimization method. This result suggests the learning mechanism of SSMs and the importance of initialization for a successful learning. (See Appendices F.1 and F.2 for further discussion).

Given that the MF is explicitly incorporated in gradient computation, it becomes possible to design models using the MF as an indicator of their memory structure. However, there is currently no established method for designing a desired MF and the constraints on the shapes of the MF remain

unclear. Because of this problem, we could not verify the important theoretical result suggesting that the memory length should be prioritized over the memory accuracies for solving universal tasks. Under the linear SSM formulation, any attempt to extend the MF has, at least in our numerical experiments, inherently led to an increase in its precision. These indicate that a long and low-precision MF is currently a theoretical requirement, but could actually exist. Therefore, the claim discussed here remains a theoretical prediction. The applicability of the MF is further discussed in Appendix C.3.

To validate the learning mechanism uncovered in this study, we also introduced a novel training approach in which the eigenvalues are fixed. As a result, approximately 10% of the total parameters were removed from the set of learnable parameters. Although the reduction is small, we found it to be sufficient to alleviate overfitting and accelerate learning. This effect stems from the inherent structure of SSMs and differs from mechanisms typically discussed in the broader machine learning literature. In addition, from the perspective of designing eigenvalues, the RC setting can also be recommended. When eigenvalues are intended to realize long memory structures, such as those derived from HiPPO, their absolute eigenvalues tend to approach 1. Because values exceeding 1 lead to output explosion, learning can make only slight adjustments to these eigenvalues, which may support the recommendation of the RC setting (see Appendix E.3). However, one scenario in which the RC setting may not be advisable is when solving tasks that require only extremely short-term memory. The reason for this was discussed at the end of section 3.2, and additional analyses should take this into account.

Although we have shown that longer memory is generally preferable, it remains an open question whether structured eigenvalues based on HiPPO (e.g., "S4Dinv" and "S4Dlin") are indeed the best choice. The reason is that although HiPPO matrices guarantee the theoretically longest memory, their indicator differs from the MF, which is inherently involved in the gradient. We confirmed that more favorable MFs can be discovered under conditions where eigenvalue learning is allowed. However, because the observed changes were modest even after convergence in accuracy and—in multiple respects—the RC setting was superior to learning, we cannot recommend the learning approach in general. Nonetheless, by developing an algorithm that explicitly extends the MF to achieve more desirable memory characteristics, better models could be realized using the newly found eigenvalues within the RC setting. We believe this study provides a theoretical foundation for future work aimed at addressing this question.

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

## A    USAGE OF LARGE LANGUAGE MODELS

We used large language models to assist non-native authors with improving the fluency and clarity of the writing. All scientific content was created and validated by the authors.

## B    MODEL

We describe the adopted model in detail here. In principle, the setting follows prior works (Gu et al., 2022b;a). To implement the SSM on modern computers, the continuous model (Eq. 1) is discretized using the zero-order hold (ZOH) assumption. The input is considered to be held constant over each sampling interval of length $\Delta$. The system can be formulated as

$$\boldsymbol{x}_{k+1} = \boldsymbol{A}\boldsymbol{x}_k + \boldsymbol{B}\boldsymbol{u}_k \qquad \boldsymbol{y}_k = \tilde{\boldsymbol{C}}\boldsymbol{x}_k + \tilde{\boldsymbol{D}}\boldsymbol{u}_k, \tag{15}$$

where $\boldsymbol{x}_k = \boldsymbol{x}(k\Delta)$, $\boldsymbol{u}_k = \boldsymbol{u}(k\Delta)$, and $\boldsymbol{y}_k = \boldsymbol{y}(k\Delta)$ denote the sampled state, input, and output at discrete time $k$, respectively. The discrete system matrices $\boldsymbol{A}$ and $\boldsymbol{B}$ are derived from the continuous-time matrices as follows: $\boldsymbol{A} = e^{\tilde{\boldsymbol{A}}\Delta}$ and $\boldsymbol{B} = \left(\int_0^\Delta e^{\tilde{\boldsymbol{A}}\tau} d\tau\right)\tilde{\boldsymbol{B}}$. Following prior work on S4 and S4D, $\Delta$ is not represented by a single fixed value within a layer but is instead assigned multiple values. For each such $\Delta$, independent SSMs are defined, and within a single layer, all of these SSMs process the input in parallel. The eigenvalues presented in Figure 1 as an example are discretized using $\Delta = 0.01$.

In section 3.1 of the main text, we used five eigenvalue realizations. We chose them because of the following reasons. The eigenvalues related to the HiPPO matrices called "S4Dinv" and "S4Dlin" are used because they were proposed in previous studies (Gu et al., 2022a) and showed great performance. The definition is as follows:

$$(\text{S4Dinv})\quad \lambda_m = -\tfrac{1}{2} + i\tfrac{N}{\pi}\left(\tfrac{N}{2m+1} - 1\right) \qquad (\text{S4Dlin})\quad \lambda_m = -\tfrac{1}{2} + i\pi m \ , \tag{16}$$

where $\lambda_m$ indicates the $m$th eigenvalue. The eigenvalues named "random" and "lin" are commonly referenced examples in earlier research on the MF. The "step" eigenvalues represent a case where the "lin" eigenvalues are extended into the complex domain because the eigenvalues of randomly generated matrices always come in complex conjugate pairs when the matrix size $N$ is even. When $N$ is odd, one real eigenvalue is added. Therefore, "step" is more suitable than "lin" for comparison with random eigenvalues.

## C    MF OF SSM

### C.1    EVALUATION

The MF of one distinct SSM is numerically calculated through the following procedure. First, using Eq. 2, we substitute the input time series $u_t$ generated by uniform random distribution and compute the state values $x_t$ over a sufficiently long duration denoted as $T$. Here, the initial $T_w$ steps are treated as a washout period. The resulting time series of states forms the matrix $\boldsymbol{X}$ whose column represents the states at each time step, and have the length of $T_w + T$. Next, for each delay $\tau$, we construct a delayed input vector $U_\tau$ of length $T$. The MFs are obtained by substituting these $\boldsymbol{X}$ and $U_\tau$ into the following formula:

$$M[u_{t-\tau}] = \frac{\boldsymbol{U}_\tau^\top \boldsymbol{X}(\boldsymbol{X}^\top\boldsymbol{X})^{-1}\boldsymbol{X}^\top\boldsymbol{U}_\tau}{\boldsymbol{U}_\tau^\top\boldsymbol{U}_\tau}, \tag{17}$$

which is equivalent to the Eq. 4 (Dambre et al., 2012). In numerical computation, due to rounding errors and noise, the MF never becomes exactly zero even if the values are already insignificant. To remove insignificant values, we used the surrogate data generated by randomly shuffling the input series in time direction. The data preserve the statistical properties of the original data while temporal dependencies can be eliminated. Replacing this data with the target series of MF $\boldsymbol{U}_\tau$, we calculated a pseudo MF to evaluate the potential error in the target MF. In practical, we calculated the pseudo MFs for 20 times by generating different surrogate data and obtain the supremum of pseudo MF by chooosing the maximum value at each $\tau$. The target MF discards values smaller than the supremum

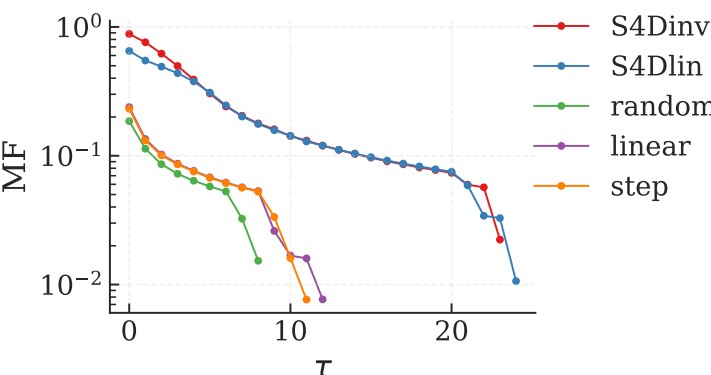

Figure 5: The average MF of five eigenvalue realizations. The horizontal axis is the delay $\tau$ of input. The system size $N$ is 32, and the results are obtained by calculating Eq. 4, where $T = 1024$.

of the pseudo MF scaled by a threshold ($= 1.2$). Counting from the delay $\tau = 0$, once elimination occurs, the MF for all subsequent $\tau$ is set to 0.

A single SSM layer is generally composed of multiple parallel SSMs. To evaluate an entire layer in terms of MF, it is necessary to take into account the MFs of all constituent SSMs. Given that, during forward propagation, the input sequence of the next layer is generated from all the output sequence of the previous, we statistically assume that all SSMs within a layer process the same input sequence. By considering the set of MFs across all SSMs in the layer, the maximum value at each delay $\tau$ can be interpreted as the maximum information processing capacity of the entire layer for one time series. Accordingly, we define this supremum of MFs as the MF representing one SSM layer. Here we additionally provide the average MF as supplementary information (Figure 5). As is evident, the indicator similarly shows that "S4Dinv" and "S4Dlin" have longer MFs compared to that of other eigenvalues.

## C.2 VANDERMONDE MATRIX

We adopted the original definition of MF (Eq. 4) instead of the analytical solution of MF (Eq. 6) because of the numerical instability of the Vandermonde matrix, which is a well studied problem (Pan, 2016; Demmel, 2000; Demmel & Koev, 2006; Drmac, 2015; Batenkov et al., 2021; Pan, 2015). In this study, the issue arises when the MF is computed using the analytical solution of MF under the eigenvalue structures of "S4Dinv" and "S4Dlin." During this computation, the MF exceeds its theoretical range of $[0, 1]$, approaching the precision limits of the machine in both positive and negative values. The computation does not produce reasonable results, suggesting that this analytical solution cannot be applied to these eigenvalues. Since the MF based on the numerical computation does not exceed its theoretical range and the actual model behavior is expected to follow the numerical procedure, we adopted the solution obtained numerically.

The issue is inherent and may arise in any computation involving $V$. As discussed in the main text, both MF and gradient calculation inevitably involve $V$. This is unavoidable when using linear RNNs, as also shown in the derivations of the original S4 (Gu et al., 2022b). However, analyzing $V$ is still beneficial. This is not only supported by previous studies demonstrating the consistency between numerical computation and the analytical solution (Guan et al., 2025), but also by the singular values explaining the long MFs of "S4Dinv" and "S4Dlin" (Figure 6). From the figure, the structured eigenvalues indicate significantly more valid singular values, explaining the results of Figure 1.

In section 3.2, the cruciality of the normalized Vandermonde matrix $V(V^\top V)^{-1}V^\top$ is revealed. To illustrate that the MF can represent matrix $V(V^\top V)^{-1}V^\top$ as an indicator, we showed the values in Figure 7. Though the matrix can be computed directly, the values obtained through this analytical solution exceed reasonable values as described in previous paragraph. Therefore, we used a numerical

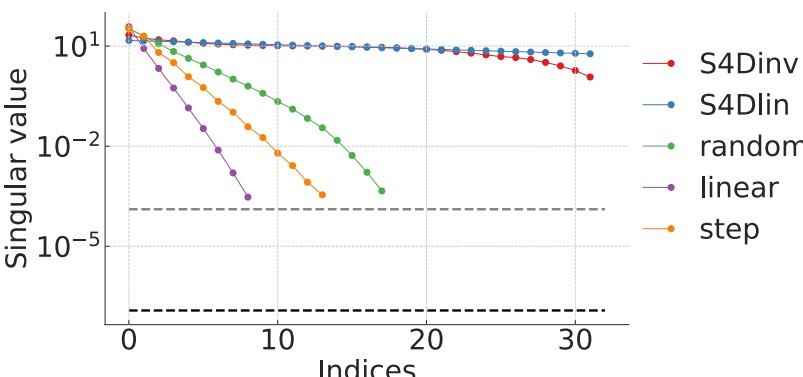

Figure 6: The singular values of matrix $\boldsymbol{V}$ calculated from five eigenvalue realizations. The horizontal axis is an index of singular values. The black line indicates the lower bound of computational accuracy $\epsilon$ determined by the common double precision environment. The calculated singular value below $\max(T, N) \times \max(\sigma) \times \epsilon$ is truncated, where $\max(\sigma)$ is the largest absolute singular values. The maximum threshold calculated from all the realizations is depicted in the gray line. The system size is $N = 32$ and $T = 1024$. The eigenvalues used here as an example are discretized using $\Delta = 0.01$.

method to calculate the matrix $\boldsymbol{V}(\boldsymbol{V}^\top \boldsymbol{V})^{-1}\boldsymbol{V}^\top$, which is based on the following analytical equation:

$$\boldsymbol{V}(\boldsymbol{V}^\top \boldsymbol{V})^{-1}\boldsymbol{V}^\top = \frac{\hat{\boldsymbol{U}}^\top \boldsymbol{X}(\boldsymbol{X}^\top \boldsymbol{X})^{-1}\boldsymbol{X}^\top \hat{\boldsymbol{U}}}{T\langle u^2 \rangle}, \tag{18}$$

where $\hat{\boldsymbol{U}} = (\boldsymbol{U}_{T-1} \quad \cdots \quad \boldsymbol{U}_0)$ and $\langle u^2 \rangle$ is the variance of $u$. The derivation of this equation follows directly from Appendix D. We calculated the right hand of the equation as numerical $\boldsymbol{V}(\boldsymbol{V}^\top \boldsymbol{V})^{-1}\boldsymbol{V}^\top$. The procedure to obtain $\boldsymbol{X}$ and $\boldsymbol{U}_\tau$ follows the method explained in the previous evaluation section

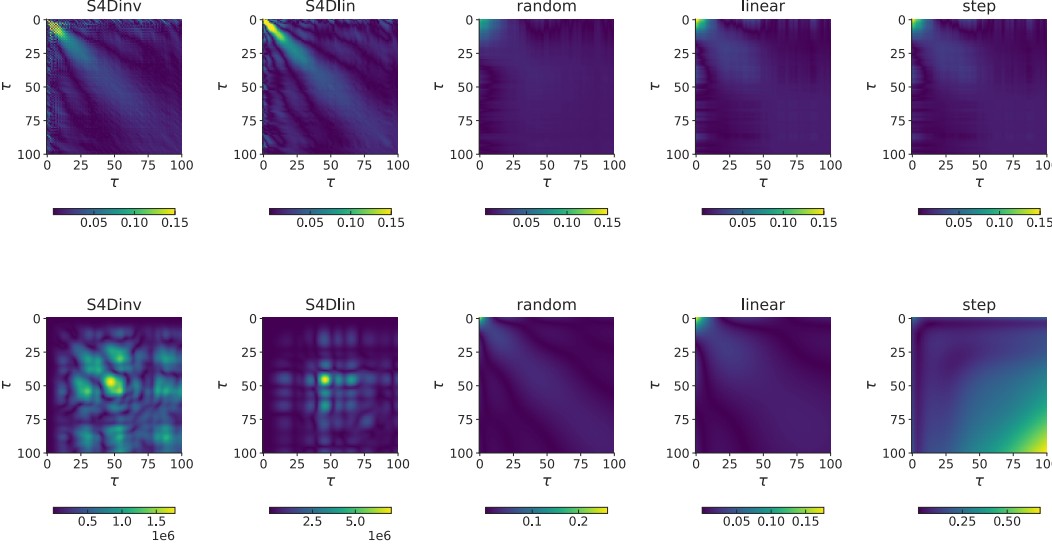

Figure 7: The value of matrix $\boldsymbol{V}(\boldsymbol{V}^\top \boldsymbol{V})^{-1}\boldsymbol{V}^\top$ for five eigenvalue realizations. The upper row shows the values following the numerical calculation, The values in the lower row are directly computed based on the formula $\boldsymbol{V}(\boldsymbol{V}^\top \boldsymbol{V})^{-1}\boldsymbol{V}^\top$. The range of colorbars in the upper row is consistent across all realizations while that of the lower row depends on the result of each realizations. Both horizontal and vertical axes indicate the delay $\tau$.

### C.3 APPLICABILITY

MF is a statistical measure for evaluating information processing and, of course, cannot exactly represent the memory of correlated inputs other than i.i.d. inputs. To justify that this assumption, which may limit the applicability of our results, is nonetheless reasonable, we performed experiments to confirm the conjectures indicated by our theoretical analysis. In the main text, we proved the theoretical equivalence between S4 and S4D from the perspective of MF as a statistical measure of information processing under i.i.d. inputs. The numerical results showed that S4D performs competitively with S4 in most tasks, suggesting that we could partially confirm the equivalence between the two models. As in prior work, however, our experiments did not demonstrate perfect equivalence, indicating that the equivalence remains purely theoretical. The theoretical results derived from MF may also not hold perfectly in an empirical sense. Nevertheless, as shown in the main text, the theoretical findings are consistent with the empirical observations and do not contradict them. Therefore, we consider that these results provide an important theoretical justification for well-known empirical findings, including the importance of initialization and the small performance gap between S4 and S4D.

## D THE PARAMETER UPDATE ALGORITHM OF STATE SPACE MODELS

Technical details for the analysis of gradient of SSMs are provided. To generally describe the equation of parameter update, we focus on the $l$th layer. Using gradient function $G$ of $\boldsymbol{y}$, the parameter update equation for the $l$th layer is defined as follows:

$$\theta_{t+1}^l = F(G, \theta_t^l, \boldsymbol{y}, \epsilon), \tag{19}$$

where $\theta^l$ is the parameter for the $l$th layer, $F$ is a function defined by a certain optimization strategy, and $\boldsymbol{y}$ is the teacher information for the entire model. The function $G = G_l \cdot G_{l+1} \cdots G_L$ is composed of the gradient function $G_k$ from each layer $k$. In this chapter, we focus on gradient $G_l$ in the $l$th layer and represent $G_l$ as $G$ by omitting the suffix $l$.

### D.1 THE DERIVATION OF THE UPDATE EQUATION WITH ARBITRARY NONLINEAR READOUT

In this section, we analytically derive the gradient $G(\boldsymbol{y})$ of one SSM layer.

**Theorem D.1.** *Gradient $G(\boldsymbol{y})$ of one SSM layer:*

$$G(\boldsymbol{y}) = \frac{2}{B} \boldsymbol{X}^\top \left[ f(\boldsymbol{X}\boldsymbol{\theta}) \odot f'(\boldsymbol{X}\boldsymbol{\theta}) - \hat{\boldsymbol{U}}^\top \boldsymbol{V} (\boldsymbol{V}^\top \boldsymbol{V})^{-1} \boldsymbol{V}^\top \bar{\boldsymbol{\alpha}} \right],$$

*where $\boldsymbol{X} \in \mathbb{R}^{T \times N}$ is a matrix whose elements consist of state series, $T$ is the time length, and $N$ is the state size. The matrix $\boldsymbol{V} \in \mathbb{R}^{T \times N}$ represents the Vandermonde matrix composed of the eigenvalues of internal weight matrix $\boldsymbol{A}$, the vector $\bar{\boldsymbol{\alpha}}$ quantifies the amount of delayed input information contained in the target time series, and $\hat{\boldsymbol{U}} = (\boldsymbol{U}_{T-1} \quad \cdots \quad \boldsymbol{U}_0)$, where vector $\boldsymbol{U}_k$ is the $k$th delayed input series. The vector $\boldsymbol{\theta}$ is the parameter of linear readout layer, $f(\cdot)$ is the activation function in the readout layer, and $B$ is an arbitrary constant.*

We begin the derivation with the formula $G(\boldsymbol{y}) = \frac{2}{T} \boldsymbol{X}^\top [(f(\boldsymbol{X}\boldsymbol{\theta}) - \boldsymbol{y}) \odot f'(\boldsymbol{X}\boldsymbol{\theta})]$. In structured state space sequence (S4) model, the state at time $t$ can be represented as:

$$\boldsymbol{x}_t = \sum_{i=1}^{t} \boldsymbol{A}^{i-1} B u_{t-i+1} = \sum_{i=1}^{t} \boldsymbol{P} \boldsymbol{\Lambda}^{i-1} \boldsymbol{P}^{-1} B u_{t-i+1}, \tag{20}$$

where initial state is $\boldsymbol{x}_0 = \boldsymbol{0}$ and $u_t$ is the input series at time $t$. We simplified $\boldsymbol{x}_t$ through the eigenvalue decomposition of internal weight matrix $\boldsymbol{A} = \boldsymbol{P}\boldsymbol{\Lambda}\boldsymbol{P}^{-1}$, where $\Lambda = \text{diag}\,(\lambda_1 \quad \lambda_2 \quad \cdots \quad \lambda_N)$, and $\lambda_m$ are the $m$th eigenvalues of $\boldsymbol{A}$. The largest absolute $\lambda_m$ is defined as the spectral radius $\rho$, which should not exceed 1 to ensure that the state value does not diverge. Defining

$$\boldsymbol{P} = (\boldsymbol{p}_1 \quad \boldsymbol{p}_2 \quad \cdots \quad \boldsymbol{p}_N) \text{ and } \boldsymbol{P}^{-1} = \begin{pmatrix} {\boldsymbol{p}_1'}^\top \\ {\boldsymbol{p}_2'}^\top \\ \vdots \\ {\boldsymbol{p}_N'}^\top \end{pmatrix},$$

$$\boldsymbol{x}_t = \boldsymbol{P} \sum_{i=1}^{t} u_{t-i+1} \mathrm{diag}\left(\lambda_1^{i-1} \quad \lambda_2^{i-1} \quad \cdots \quad \lambda_N^{i-1}\right) \begin{pmatrix} {\boldsymbol{p}_1'}^\top \boldsymbol{B} \\ {\boldsymbol{p}_2'}^\top \boldsymbol{B} \\ \vdots \\ {\boldsymbol{p}_N'}^\top \boldsymbol{B} \end{pmatrix} \tag{21}$$

$$= \boldsymbol{P} \sum_{i=1}^{t} \begin{pmatrix} {\boldsymbol{p}_1'}^\top \boldsymbol{B} \lambda_1^{i-1} \\ {\boldsymbol{p}_2'}^\top \boldsymbol{B} \lambda_2^{i-1} \\ \vdots \\ {\boldsymbol{p}_N'}^\top \boldsymbol{B} \lambda_N^{i-1} \end{pmatrix} u_{t-i+1}. \tag{22}$$

Considering the $k$th delayed state $\boldsymbol{x}_{t-k}$, the formula is given by

$$\boldsymbol{x}_{t-k} = \boldsymbol{P} \begin{pmatrix} p_1'_{\boldsymbol{B}} \boldsymbol{\Lambda}_1^\top \boldsymbol{U}_k \\ p_2'_{\boldsymbol{B}} \boldsymbol{\Lambda}_2^\top \boldsymbol{U}_k \\ \vdots \\ p_N'_{\boldsymbol{B}} \boldsymbol{\Lambda}_N^\top \boldsymbol{U}_k \end{pmatrix} = \boldsymbol{Q} \begin{pmatrix} \boldsymbol{\Lambda}_1^\top \boldsymbol{U}_k \\ \boldsymbol{\Lambda}_2^\top \boldsymbol{U}_k \\ \vdots \\ \boldsymbol{\Lambda}_N^\top \boldsymbol{U}_k \end{pmatrix},$$

where $\boldsymbol{U}_k = \left(u_{t-k-(T-1)} \quad \cdots u_{t-k-(1)} \quad u_{t-k-(0)}\right)^\top$, $\boldsymbol{\Lambda}_m = \left(\lambda_m^{T-1} \quad \cdots \quad \lambda_m^{1} \quad \lambda_m^{0}\right)^\top$, and $\boldsymbol{Q} = \left({\boldsymbol{p}_1'}^\top \boldsymbol{B} \boldsymbol{p}_1 \quad {\boldsymbol{p}_2'}^\top \boldsymbol{B} \boldsymbol{p}_2 \quad \cdots \quad {\boldsymbol{p}_N'}^\top \boldsymbol{B} \boldsymbol{p}_N\right)$. As a result, $\boldsymbol{X}$ is described as follows:

$$\boldsymbol{X}^\top = \boldsymbol{Q} \begin{pmatrix} \boldsymbol{\Lambda}_1^\top \boldsymbol{U}_{T-1} & & \boldsymbol{\Lambda}_1^\top \boldsymbol{U}_1 & \boldsymbol{\Lambda}_1^\top \boldsymbol{U}_0 \\ \boldsymbol{\Lambda}_2^\top \boldsymbol{U}_{T-1} & & \boldsymbol{\Lambda}_2^\top \boldsymbol{U}_1 & \boldsymbol{\Lambda}_2^\top \boldsymbol{U}_0 \\ \vdots & \cdots & \vdots & \vdots \\ \boldsymbol{\Lambda}_N^\top \boldsymbol{U}_{T-1} & & \boldsymbol{\Lambda}_N^\top \boldsymbol{U}_1 & \boldsymbol{\Lambda}_N^\top \boldsymbol{U}_0 \end{pmatrix} = \boldsymbol{Q} \begin{pmatrix} \boldsymbol{\Lambda}_1^\top \\ \boldsymbol{\Lambda}_2^\top \\ \vdots \\ \boldsymbol{\Lambda}_N^\top \end{pmatrix} (\boldsymbol{U}_{T-1} \quad \cdots \quad \boldsymbol{U}_1 \quad \boldsymbol{U}_0). \tag{23}$$

Here, we go back the formula of the gradient:

$$G(\boldsymbol{y}) = \frac{2}{B} \boldsymbol{X}^\top \left[f(\boldsymbol{X}\boldsymbol{\theta}) \odot f'(\boldsymbol{X}\boldsymbol{\theta}) - \bar{\boldsymbol{y}}\right] \tag{24}$$

$$= \frac{2}{B} \boldsymbol{X}^\top \left[f(\boldsymbol{X}\boldsymbol{\theta}) \odot f'(\boldsymbol{X}\boldsymbol{\theta}) - \boldsymbol{X}(\boldsymbol{X}^\top \boldsymbol{X})^{-1} \boldsymbol{X}^\top \bar{\boldsymbol{y}}\right], \tag{25}$$

where $\bar{\boldsymbol{y}} = \boldsymbol{y} \odot f'(\boldsymbol{X}\boldsymbol{\theta})$ represents the label modified by the derivative of the activation function. The covariance matrix $\boldsymbol{X}^\top \boldsymbol{X}$ has the following form:

$$\boldsymbol{X}^\top \boldsymbol{X} = \boldsymbol{Q} \boldsymbol{V}^\top (\boldsymbol{U}_{T-1} \quad \cdots \quad \boldsymbol{U}_0)(\boldsymbol{U}_{T-1} \quad \cdots \quad \boldsymbol{U}_0)^\top \boldsymbol{V} \boldsymbol{Q}^\top, \tag{26}$$

where $\boldsymbol{V} = \begin{pmatrix} \lambda_1^{T-1} & \lambda_1^{T-2} & \cdots & \lambda_1^{1} & 1 \\ \lambda_2^{T-1} & \lambda_2^{T-2} & \cdots & \lambda_2^{1} & 1 \\ \vdots & & & & \\ \lambda_N^{T-1} & \lambda_N^{T-2} & \cdots & \lambda_N^{1} & 1 \end{pmatrix}^\top$. Similar to the MF, to evaluate the inher-

ent statistical structure, we assume that the input $u_t$ is an independent and identically distributed random variable, suggesting that delayed time series $\boldsymbol{U}_k$ are orthogonal to each other. Using $(\boldsymbol{U}_{T-1} \quad \cdots \quad \boldsymbol{U}_0) = (\boldsymbol{U}_{T-1} \quad \cdots \quad \boldsymbol{U}_0)^\top$, the covariance matrix $\boldsymbol{X}^\top \boldsymbol{X}$ reduces to the following form:

$$\boldsymbol{X}^\top \boldsymbol{X} = \boldsymbol{Q} \boldsymbol{V}^\top \begin{pmatrix} {\boldsymbol{U}_{T-1}}^\top \\ \vdots \\ {\boldsymbol{U}_0}^\top \end{pmatrix} (\boldsymbol{U}_{T-1} \quad \cdots \quad \boldsymbol{U}_0) \boldsymbol{V} \boldsymbol{Q}^\top = T\langle u^2 \rangle \boldsymbol{Q} \boldsymbol{V}^\top \boldsymbol{V} \boldsymbol{Q}^\top, \tag{27}$$

where $\langle u^2 \rangle$ is the variance of $u$. Since

$$\boldsymbol{X}(\boldsymbol{X}^\top \boldsymbol{X})^{-1}\boldsymbol{X}^\top = (T\langle u^2 \rangle)^{-1} \begin{pmatrix} \boldsymbol{U}_{T-1} & \cdots & \boldsymbol{U}_0 \end{pmatrix}^\top \boldsymbol{V}(\boldsymbol{V}^\top \boldsymbol{V})^{-1}\boldsymbol{V}^\top \begin{pmatrix} \boldsymbol{U}_{T-1} & \cdots & \boldsymbol{U}_0 \end{pmatrix}, \quad (28)$$

we could observe that the gradient of SSM (Eq. 25) is determined by the matrix $\boldsymbol{V}(\boldsymbol{V}^\top \boldsymbol{V})^{-1}\boldsymbol{V}^\top$. As a result, the gradient $G(\boldsymbol{y})$ is described by

$$G(\boldsymbol{y}) = \frac{2}{B}\boldsymbol{X}^\top \left[ f(\boldsymbol{X}\boldsymbol{\theta}) \odot f'(\boldsymbol{X}\boldsymbol{\theta}) - \frac{1}{T\langle u^2 \rangle}\hat{\boldsymbol{U}}^\top \boldsymbol{V}(\boldsymbol{V}^\top \boldsymbol{V})^{-1}\boldsymbol{V}^\top \hat{\boldsymbol{U}}^\top \bar{\boldsymbol{y}} \right], \quad (29)$$

where $\hat{\boldsymbol{U}} = \begin{pmatrix} \boldsymbol{U}_{T-1} & \cdots & \boldsymbol{U}_0 \end{pmatrix}$. Using the fact that linear SSM only performs linear information processing, all possible output of the SSM can be described as the linear combination of past inputs series $\bar{\boldsymbol{y}} = \sum_{k=0}^{T-1} \bar{\alpha}_k \boldsymbol{U}_k = \hat{\boldsymbol{U}}\bar{\boldsymbol{\alpha}}$, where $\{\bar{\alpha}_\tau\}_{\tau=0}^{T-1}$ is a certain series characterizing $\bar{\boldsymbol{y}}$ and $\bar{\boldsymbol{\alpha}} = \begin{pmatrix} \bar{\alpha}_{T-1} & \cdots & \bar{\alpha}_0 \end{pmatrix}^\top$. By substituting this $\bar{\boldsymbol{y}}$ into $G(\boldsymbol{y})$, the gradient is computed as

$$G(\boldsymbol{y}) = \frac{2}{B}\boldsymbol{X}^\top \left[ f(\boldsymbol{X}\boldsymbol{\theta}) \odot f'(\boldsymbol{X}\boldsymbol{\theta}) - \frac{1}{T\langle u^2 \rangle}\hat{\boldsymbol{U}}^\top \boldsymbol{V}(\boldsymbol{V}^\top \boldsymbol{V})^{-1}\boldsymbol{V}^\top \hat{\boldsymbol{U}}^\top \hat{\boldsymbol{U}}\bar{\boldsymbol{\alpha}} \right] \quad (30)$$

$$= \frac{2}{B}\boldsymbol{X}^\top \left[ f(\boldsymbol{X}\boldsymbol{\theta}) \odot f'(\boldsymbol{X}\boldsymbol{\theta}) - \hat{\boldsymbol{U}}^\top \boldsymbol{V}(\boldsymbol{V}^\top \boldsymbol{V})^{-1}\boldsymbol{V}^\top \bar{\boldsymbol{\alpha}} \right]. \quad (31)$$

### D.2 THE UPDATE EQUATION WITH LINEAR ACTIVATION FUNCTION

Next, we consider the linear activation case in order to simplify the problem and to obtain a more intuitive and straightforward understanding.

**Lemma D.2.** *Gradient $G(\boldsymbol{y})$ of a linear SSM layer:*

$$G(\boldsymbol{y}) = 2\langle u^2 \rangle B \boldsymbol{V}^\top (B\boldsymbol{V}\boldsymbol{\theta} - \boldsymbol{V}(\boldsymbol{V}^\top \boldsymbol{V})^{-1}\boldsymbol{V}^\top \boldsymbol{\alpha}),$$

*where $\boldsymbol{V} \in \mathbb{R}^{T \times N}$ is the Vandermonde matrix composed of the eigenvalues of internal weight matrix $\boldsymbol{A}$, $T$ is the time length, and $N$ is the states size. The vector $\boldsymbol{\alpha}$ quantifies the amount of delayed input information contained in the target time series, $\langle u^2 \rangle$ is the variance of input $u$, and $B$ is an arbitrary constant.*

We focus on the gradient before the nonlinear activation layer, where gradient is described by

$$G(\boldsymbol{y}) = \frac{2}{T}\boldsymbol{X}^\top (\boldsymbol{X}\boldsymbol{\theta} - \boldsymbol{y}). \quad (32)$$

Following the main text, we consider a typical task, where $\boldsymbol{y}$ is the delayed input series $\boldsymbol{U}_\tau \in \mathbb{R}^T$,

$$G(\boldsymbol{U}_\tau) = \frac{2}{T}(\boldsymbol{X}^\top \boldsymbol{X}\boldsymbol{\theta} - \boldsymbol{X}^\top \boldsymbol{U}_\tau), \quad (33)$$

where $\tau$ is the delay. Because

$$\boldsymbol{X}^\top \boldsymbol{U}_\tau = \boldsymbol{Q}\boldsymbol{V}^\top \begin{pmatrix} \boldsymbol{U}_{T-1}^\top \\ \vdots \\ \boldsymbol{U}_0^\top \end{pmatrix} \boldsymbol{U}_\tau = \boldsymbol{Q}\boldsymbol{V}^\top \begin{pmatrix} 0 \\ \vdots \\ 0 \\ \boldsymbol{U}_\tau^\top \boldsymbol{U}_\tau \\ 0 \\ \vdots \\ 0 \end{pmatrix} = T\langle u^2 \rangle \boldsymbol{Q}\boldsymbol{V}_\tau, \quad (34)$$

where $\boldsymbol{V}_\tau = \begin{pmatrix} \lambda_1^\tau & \lambda_2^\tau & \cdots & \lambda_N^\tau \end{pmatrix}^\top$, combining the result of nonlinear activation case, we obtain

$$G(\boldsymbol{U}_\tau) = \frac{2}{T}(T\langle u^2 \rangle \boldsymbol{Q}\boldsymbol{V}^\top \boldsymbol{V}\boldsymbol{Q}^\top \boldsymbol{\theta} - T\langle u^2 \rangle \boldsymbol{Q}\boldsymbol{V}_\tau) = 2\langle u^2 \rangle (\boldsymbol{Q}\boldsymbol{V}^\top \boldsymbol{V}\boldsymbol{Q}^\top \boldsymbol{\theta} - \boldsymbol{Q}\boldsymbol{V}_\tau), \quad (35)$$

Following the MF framework introduced in the main text, we simplify S4 to S4D, indicating that $\boldsymbol{P} = \boldsymbol{E}$ and $\boldsymbol{Q} = \text{diag}(\boldsymbol{B})$. Using $\boldsymbol{B} = B\mathbf{1}$, we can compute the gradient as

$$G(\boldsymbol{U}_\tau) = 2\langle u^2 \rangle \boldsymbol{V}^\top \boldsymbol{V}(B^2 \boldsymbol{\theta} - B(\boldsymbol{V}^\top \boldsymbol{V})^{-1}\boldsymbol{V}_\tau) \quad (36)$$

$$= 2\langle u^2 \rangle B \boldsymbol{V}^\top (B\boldsymbol{V}\boldsymbol{\theta} - \boldsymbol{V}(\boldsymbol{V}^\top \boldsymbol{V})^{-1}\boldsymbol{V}_\tau). \quad (37)$$

All the possible gradients can be obtained by substituting the output $\boldsymbol{y} = \sum_{k=0}^{T-1} \alpha_k \boldsymbol{U}_k$ because the linear combination of the past inputs series fully describes the output of the SSM, where $\{\alpha_\tau\}_{\tau=0}^{T-1}$ is a certain series characterizing $\boldsymbol{y}$.

$$G(\boldsymbol{y}) \quad = \quad 2\langle u^2 \rangle \boldsymbol{V}^\top \boldsymbol{V} (B^2 \boldsymbol{\theta} - B \sum_{k=0}^{T-1} \alpha_k (\boldsymbol{V}^\top \boldsymbol{V})^{-1} \boldsymbol{V}_k) \tag{38}$$

$$= \quad 2\langle u^2 \rangle B \boldsymbol{V}^\top (B \boldsymbol{V} \boldsymbol{\theta} - \boldsymbol{V}(\boldsymbol{V}^\top \boldsymbol{V})^{-1} \boldsymbol{V}^\top \boldsymbol{\alpha}). \tag{39}$$

As discussed in Section 3.2, this Lemma D.2 can sufficiently explain the gradient of a SSM layer because the input of the subsequent layer in the forward propagation completely depends on the output of this linear SSM layer. We can assume the existence of an ideal outputs of the current SSM layer independent from other layers.

### D.3 THE IMPORTANCE OF MF

For example, we consider SGD. The terminal parameter $\boldsymbol{\theta}$ can be analytically derived. $\boldsymbol{\theta}$ is obtained by solving the equation

$$\boldsymbol{\theta} \quad = \quad \boldsymbol{\theta} - \eta_t 2\langle u^2 \rangle (B^2 \boldsymbol{V}^\top \boldsymbol{V} \boldsymbol{\theta} - B \boldsymbol{V}^\top \boldsymbol{\alpha}). \tag{40}$$

As a result, we obtain $\boldsymbol{\theta}$ and the output $\boldsymbol{X}\boldsymbol{\theta}$ as follows:

$$\boldsymbol{\theta} \quad = \quad \frac{1}{B}(\boldsymbol{V}^\top \boldsymbol{V})^{-1} \boldsymbol{V}^\top \boldsymbol{\alpha}, \tag{41}$$

$$\boldsymbol{X}\boldsymbol{\theta} \quad = \quad \begin{pmatrix} \boldsymbol{U}_{T-1} & \cdots & \boldsymbol{U}_0 \end{pmatrix}^\top \boldsymbol{V}^\top (\boldsymbol{V}\boldsymbol{V}^\top)^{-1} \boldsymbol{V} \boldsymbol{\alpha}, \tag{42}$$

where matrix $\boldsymbol{V}(\boldsymbol{V}^\top \boldsymbol{V})^{-1} \boldsymbol{V}^\top$ appears again, suggesting that MF is inherent in the update algorithm. Actually, this is a reasonable result, as both SGD and MF are derived using the MSE as their loss function. The result obtained here proves the equivalence of the two optimization strategies in a statistical view.

The role of MF can be further revealed from Eqs. 12 and 42, especially in the following part:

$$\boldsymbol{V}(\boldsymbol{V}^\top \boldsymbol{V})^{-1} \boldsymbol{V}^\top \boldsymbol{\alpha}, \tag{43}$$

as $\boldsymbol{\alpha}$ is itself the MF of $\boldsymbol{y}$. This follows directly from Eq. 17 by replacing $\boldsymbol{X}$ with $\boldsymbol{y}$ when $u$ is regarded as the source of $\boldsymbol{y}$. This is in fact straightforward given the definition of $\alpha$. The result proposes that the gradient of SSM is statistically characterized by the product of the two MFs: MF of the SSM itself and MF of the teacher information $\boldsymbol{y}$.

### D.4 INTERPRETATION OF THEORETICAL RESULTS

To interpret the meaning of gradient, we define a successful parameter update.

**Definition D.3.** *Successful parameter update. We say that parameter update is successful if the parameter update rule $F$ (Eq. 19) depends on teacher information. In this paper, the teacher information is represented by $\alpha$ in Lemma D.2.*

We use this term if the norm of $\boldsymbol{V}(\boldsymbol{V}^\top \boldsymbol{V})^{-1} \boldsymbol{V}^\top \boldsymbol{\alpha}$ is larger than 0. According to this definition and the gradient formula, the learning mechanism of SSM could be partly revealed.

**Theorem D.4.** *In SSM, gradient-based optimization has no explicit mechanism to recover from the loss of teacher information.*

*Proof.* To begin with, we assume that the loss of teacher information $\boldsymbol{\alpha} = \begin{pmatrix} \alpha_0 & \alpha_1 & \cdots & \alpha_{T-1} \end{pmatrix}^\top$ occurs during the gradient update at the current step $t$, where $\alpha_\tau$ is the magnitude of the $\tau$th delayed inputs in the label. For simplicity, we focus on the input of one delay $k$ and assumes $|\alpha_k| > 0$. The teacher information is modified to $\boldsymbol{V}(\boldsymbol{V}^\top \boldsymbol{V})^{-1} \boldsymbol{V}^\top \boldsymbol{\alpha}$ as explained in Eq. 12. The loss of teacher information $\alpha_k$ is described as

$$Y(\{\alpha_{\tau \neq k}\}_{\tau=0}^{T-1}) = Y(\alpha_0, \alpha_1, \cdots, \alpha_{k-1}, \alpha_{k+1}, \cdots, \alpha_{T-1}) \tag{44}$$

$$= \boldsymbol{V}(\boldsymbol{V}^\top \boldsymbol{V})^{-1} \boldsymbol{V}^\top \boldsymbol{\alpha}, \tag{45}$$

suggesting that the teacher information become independent from $\alpha_k$ and we can define an arbitrary value for $\alpha_k$. This loss may occur because the matrix $\boldsymbol{V}(\boldsymbol{V}^\top\boldsymbol{V})^{-1}\boldsymbol{V}^\top$ has the role to screen the information associated to a certain delay.

To assume this condition, we prove the existence of the Eq. 45. Using the fact that $\alpha_\tau$ is multiplied by the value representing how much the information about the input with delay $\tau$ is lost and that the decaying shape of the matrix is represented by the MF $M[u_{t-\tau}]$, the linear product $\boldsymbol{V}(\boldsymbol{V}^\top\boldsymbol{V})^{-1}\boldsymbol{V}^\top\boldsymbol{\alpha}$ is described as

$$\boldsymbol{V}(\boldsymbol{V}^\top\boldsymbol{V})^{-1}\boldsymbol{V}^\top\boldsymbol{\alpha} = Y(\{M[u_{t-\tau}]\cdot\alpha_\tau\}_{\tau=0}^{T-1}). \tag{46}$$

If the MF indicates $M[u_{t-k}] = 0$,

$$M[u_{t-\tau}]\cdot\alpha_k = 0. \tag{47}$$

Therefore, we can discard $\alpha_k$ as an argument of the function $Y$:

$$\boldsymbol{V}(\boldsymbol{V}^\top\boldsymbol{V})^{-1}\boldsymbol{V}^\top\boldsymbol{\alpha} = Y(\{\alpha_{\tau\neq k}\}_{\tau=0}^{T-1}), \tag{48}$$

proving that the loss of teacher information can occur.

Following this assumption (Eq. 45), the gradient of the SSM layer is described as

$$G(\boldsymbol{y}) = 2\langle u^2\rangle B\boldsymbol{V}^\top(B\boldsymbol{V}\boldsymbol{\theta} - Y(\{\alpha_{\tau\neq k}\}_{\tau=0}^{T-1})) \tag{49}$$

$$= G(\{\alpha_{\tau\neq k}\}_{\tau=0}^{T-1}). \tag{50}$$

The gradient becomes independent from the label information associated with the $k$th delayed input, suggesting that successful parameter update will not proceed at all regarding the label information associated with the delay. The independence also proposes that no mechanism exists to reflect the fact that the information has been lost at the current step $t$. Therefore, this failure of losing teacher information will not be improved in the next step $t+1$. Recursively, the loss of information inevitably persists in later steps. □

This result shows that, at initialization, the internal weight $\mathbf{A}$ must be appropriately designed to ensure a sufficient memory, expressed as $\mathbf{V}(\mathbf{V}^\top\mathbf{V})^{-1}\mathbf{V}^\top$. Subsequently, the condition to solve an arbitrary task is derived.

**Lemma D.5.** *SSM requires an infinitely long memory to solve an arbitrary task.*

*Proof.* Here we consider a task characterized by teacher information $\alpha_k$ whose absolute values $|\alpha_k|$ are all larger than 0, where $k = 0, 1, 2, \cdots, T-1$ and theoretically, $T \to \infty$. To ensure a successful parameter update to solve this task, we must initialize the weight with an internal matrix $\boldsymbol{A}$ that meets the following condition: the values of $\boldsymbol{V}(\boldsymbol{V}^\top\boldsymbol{V})^{-1}\boldsymbol{V}^\top$ remain sufficiently larger than 0 in the distant past such that all $\alpha_k$ are not 0. Because the decay of $\boldsymbol{V}(\boldsymbol{V}^\top\boldsymbol{V})^{-1}\boldsymbol{V}^\top$ is represented by the MF, the MF should be infinitely long to solve this task. □

**Theorem D.6.** *Memory length takes priority over memory precision for a successful parameter update.*

*Proof.* According to Lemma D.5, an infinitely long memory is required to solve an arbitrary task. Combining this fact with the tradeoff between the accuracy and length of memory (Eq. 5), the accuracy of memory should be abandoned. □

According to Theorem D.4, two conjectures can be derived by focusing on the memory structure when full parameters are learned. One conjecture is formulated as follows.

**Conjecture D.7.** *Memory decreases as learning proceeds. The decrease of memory is defined as*

$$M[u_{t-\tau}]_k > M[u_{t-\tau}]_{k+1}, \tag{51}$$

*where $k$ indicates the number of step for parameter updates. This occurs because the memory architecture can be adjusted in accordance with the capacity required by the task.*

To formula another conjecture, we define the word "unintended extension of memory".

**Definition D.8.** *Unintended extension of memory. Assume that the input information $u_{t-\tau_1}$ is necessary in order to predict the label information $y_t$ in a task, and that, for $u_{t-\tau_1}$, the MF satisfies $M[u_{t-\tau_1}] > 0$. Then, consider the MF for a more distant past input $u_{t-\tau_2}$ with $\tau_2 > \tau_1$, i.e., $M[u_{t-\tau_2}] \geq 0$. Unintended extension refers to the situation where, during training, the model tries to improve $M[u_{t-\tau_1}]$, and as a side effect, $M[u_{t-\tau_2}]$ also increases.*

Following this definition, second conjecture is formulated.

**Conjecture D.9.** *Unintended extension of memory occurs as learning proceeds.*

For completeness, we note that no explicit memory extension occurs because of Theorem D.4.

The two conjectures D.7 and D.9 can coexist, but the problem is that the former is beneficial for learning, whereas the latter is detrimental. The necessity of conducting experiments in the RC setting arises from these two opposing conjectures to verify which predicted phenomenon surpass the other. Specifically, regarding the unintended extension of memory, we focus on a particularly important case where, $M[u_{t-\tau_2}] = 0$ before training. We empirically confirm in Fig. 3 that, in this case, unintended extension indeed occurs (Section 3.3).

# E    EXPERIMENTAL DETAILS

In this section, we provide the detailed methods and additional results for the experiments.

## E.1    TASK

The task of LRA benchmark includes long listops (Listops), byte-level text classification (Text), byte-level document retrieval (Retrieval), image classification on sequences of pixels (Image), a task to determine from pixel sequences whether a valid path exists between two points (Pathfinder), and another more difficult Pathfinder task with longer sequence (PathX).

## E.2    HYPERPARAMETERS AND ENVIRONMENT

The model are constructed following the official GitHub repository[1] provided by the authors of S4 (Gu et al., 2022b). According to the result that S4 and S4D are equivalent, we focus on the S4D model which has lower computational cost. The hyperparameters are described in the Table 2. `n_layers`, `d_model`, and `n_ssm` denote the number of layers, the number of SSMs per layer, and the number of distinct matrices $\boldsymbol{A}$, respectively. `lr`, `lr_layer`, and `lr_dt`, indicate the learning rate for the entire model, the learning rate for each layer, and the learning rate for $\Delta$, respectively. `dt_min` and `dt_max` correspond to the minimum and maximum values of $\Delta$, respectively. In the implementation, $N$ is defined by `d_state`/2, which produces $N = 32$ as specified in the main text. The activation function is defined as $f = $ GELU. In practical, a bidirectional operation was introduced. With this operation, the SSM kernel outputs two time series: the sequence corresponding to the original ordering of timesteps, and an additional time-reversed sequence. As a result, the linear readout, $\boldsymbol{W}^{(\ell)}$ and $\boldsymbol{b}^{(\ell)}$, receives an input sequence that is twice the length of the layer input time series. In addition, after the linear readout layer, post activation $g = $ GLU was introduced. Cross-entropy is employed as the loss function in the final layer. Our results confirm that the findings are consistent with the theoretical outcomes obtained using MSE.

The computations were performed using two NVIDIA A100 Tensor Core GPU [40GB] and CPU with 4 threads. The maximum memory usage of the CPU was approximately 20 GB to run the training. Excluding, PathX, each training and evaluation session was completed in approximately five hours. For PathX, it took about twenty hours.

## E.3    ADDITIONAL RESULTS

In Figure 3, we showed the supremum MF of one SSM layer when the training of eigenvalues is allowed. To demonstrate completeness, we additionally provide the average MF before and after

---

[1]`https://github.com/state-spaces/s4/tree/simple`

Table 2: Hyperparameters of the model

| parameter | Listops | Text | Retrieval | Image | Pathfinder | PathX |
|---|---|---|---|---|---|---|
| n_layers | 8 | 6 | 6 | 6 | 6 | 6 |
| d_model | 128 | 256 | 256 | 512 | 256 | 256 |
| d_state | 64 | 64 | 64 | 64 | 64 | 64 |
| n_ssm | 128 | 256 | 256 | 2 | 256 | 256 |
| lr | 0.01 | 0.01 | 0.01 | 0.01 | 0.004 | 0.0005 |
| lr_layer | 0.001 | 0.001 | 0.001 | 0.001 | 0.001 | 0.0005 |
| lr_dt | 0.01 | 0.01 | 0.01 | 0.001 | 0.001 | 0.0005 |
| dt_min | 0.001 | 0.001 | 0.001 | 0.001 | 0.001 | 0.0001 |
| dt_max | 0.1 | 0.1 | 0.1 | 0.1 | 0.1 | 0.01 |
| dropout | 0. | 0. | 0. | 0.1 | 0. | 0. |
| weight_decay | 0.05 | 0.05 | 0.05 | 0.05 | 0.03 | 0.05 |
| batch_size | 50 | 16 | 32 | 50 | 64 | 16 |
| max_eopch | 40 | 32 | 20 | 200 | 200 | 50 |

training (Figure 9). As with the supremum MF, the MF for the structured eigenvalues "S4Dinv" and "S4Dlin" exhibits longer memory compared to other eigenvalues. To reveal the reason of the modest change in the MFs, we show the eigenvalues before and after training (Figure 8). From the initial setting, the absolute eigenvalues are almost 1, and this characteristic is maintained even after training.

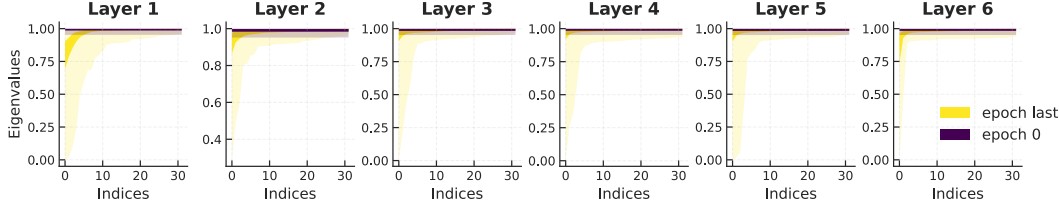

Figure 8: The eigenvalues of five realizations before and after training. The horizontal axis is the index. Color indicates the epoch. The dark shading represents the interquartile range between Q1 and Q3, while the light shading indicates the range from the minimum to the maximum. The system size $N$ is 32. As an example, the case of Pathfinder task is shown.

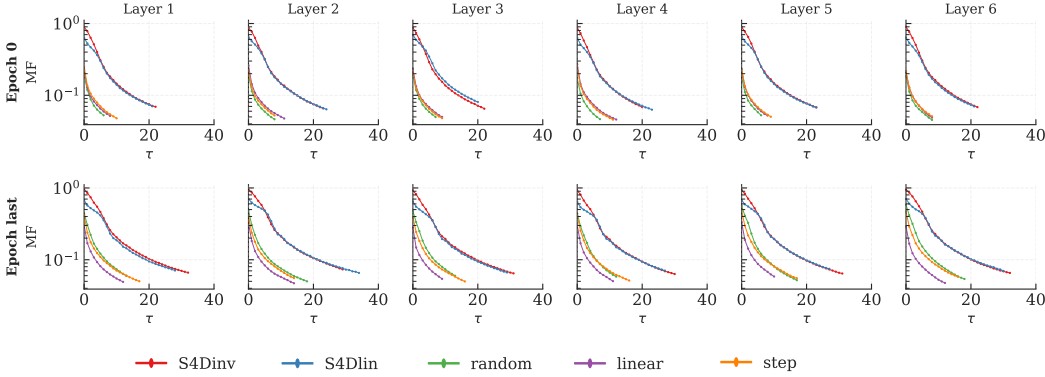

Figure 9: The MFs of each layer before and after training. The upper (lower) panels show MFs before (after) training. From left to right, the column indicates the each layer. The horizontal (vertical) axes are delay $\tau$ of input (MF). The five colors indicates the eigenvalue realizations. The state size $N$ is 32, and the results are obtained by calculating Eq. 4, where $T = 1024$. As an example, we show the case of Pathfinder task.

Furthermore, the loss and accuracy during the training and test phase are provided as supplementary materials.

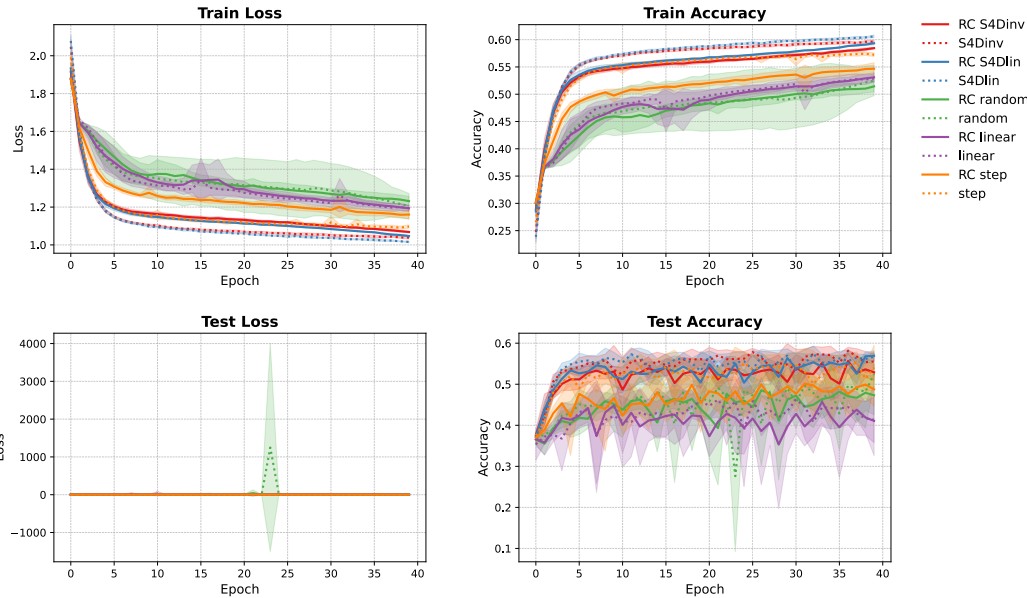

Figure 10: Test loss and accuracy of during the learning of Listops task. The upper (lower) row indicates the metrics of train (test). The column corresponds to each metric. The horizontal axes are epochs. The colors indicate eigenvalue realizations. The solid lines are the model under RC setting while the dotted lines are the model whose eigenvalues are allowed to learn.

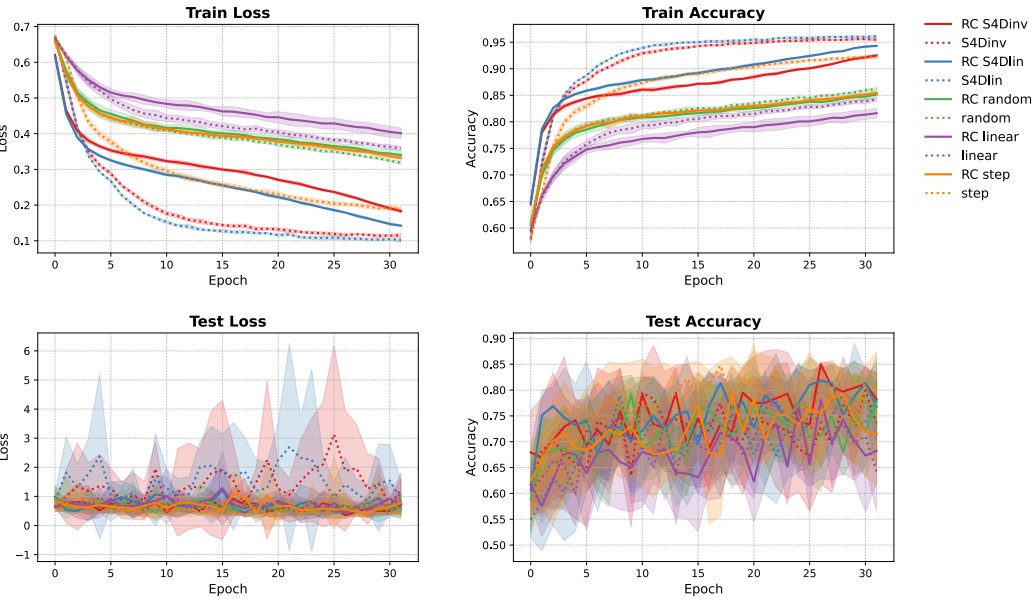

Figure 11: Test loss and accuracy of during the learning of Text task. The upper (lower) row indicates the metrics of train (test). The column corresponds to each metric. The horizontal axes are epochs. The colors indicate eigenvalue realizations. The solid lines are the model under RC setting while the dotted lines are the model whose eigenvalues are allowed to learn.

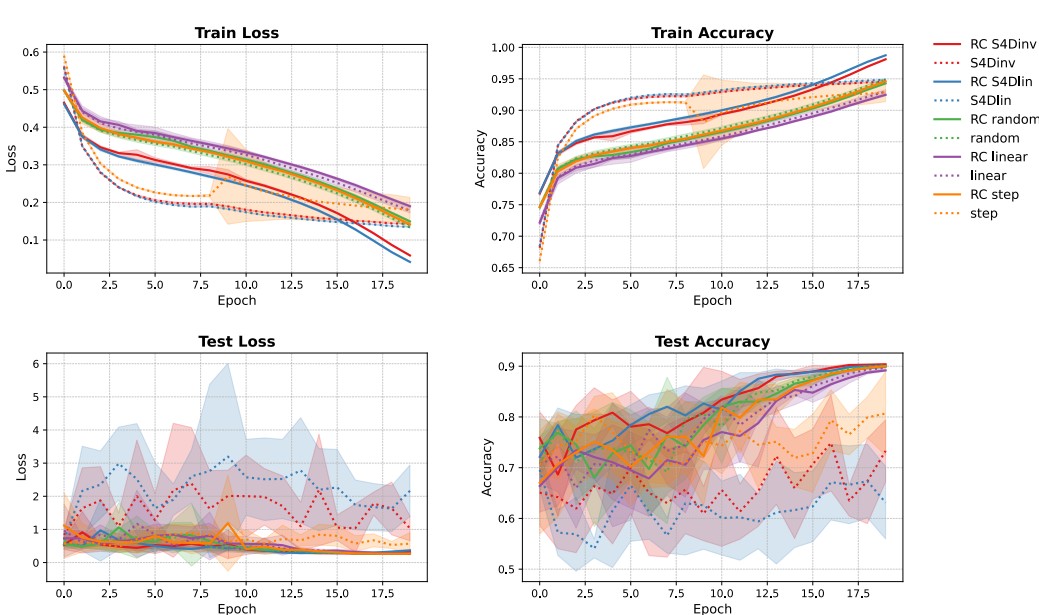

Figure 12: Test loss and accuracy of during the learning of Retrieval task. The upper (lower) row indicates the metrics of train (test). The column corresponds to each metric. The horizontal axes are epochs. The colors indicate eigenvalue realizations. The solid lines are the model under RC setting while the dotted lines are the model whose eigenvalues are allowed to learn.

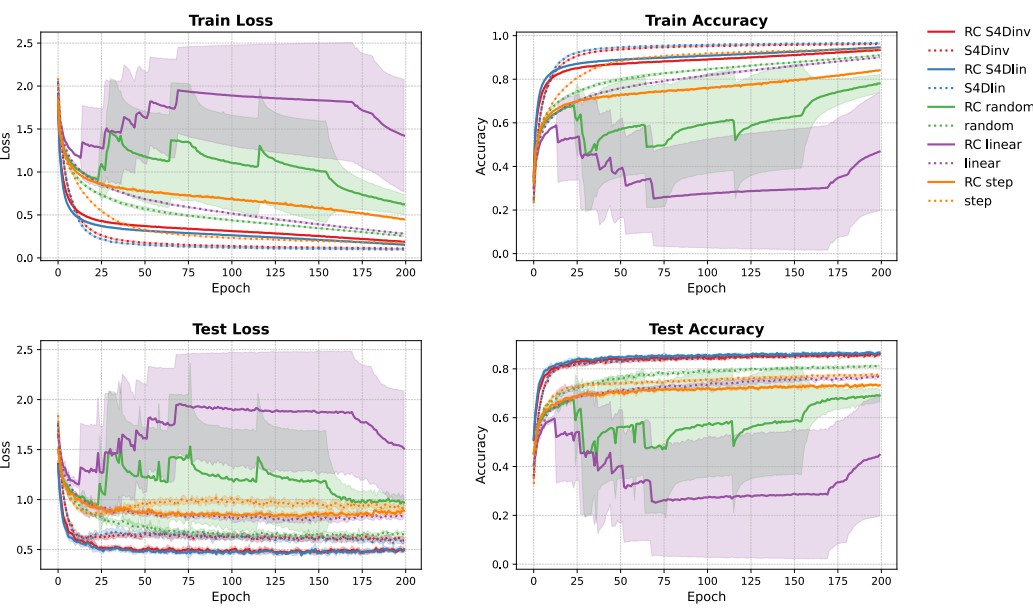

Figure 13: Test loss and accuracy of during the learning of Image task. The upper (lower) row indicates the metrics of train (test). The column corresponds to each metric. The horizontal axes are epochs. The colors indicate eigenvalue realizations. The solid lines are the model under RC setting while the dotted lines are the model whose eigenvalues are allowed to learn.

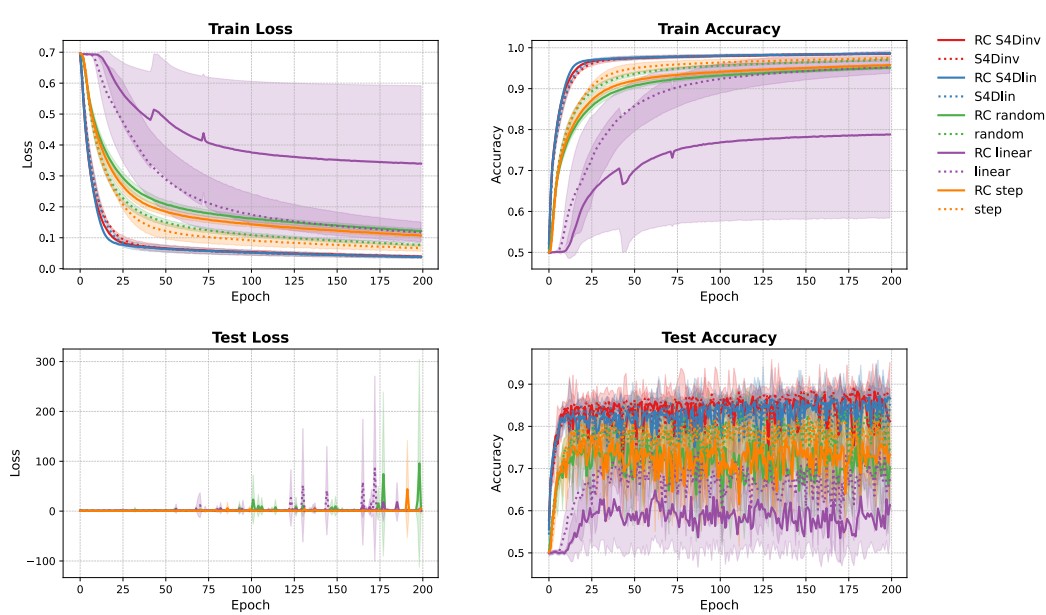

Figure 14: Test loss and accuracy of during the learning of Pathfinder task. The upper (lower) row indicates the metrics of train (test). The column corresponds to each metric. The horizontal axes are epochs. The colors indicate eigenvalue realizations. The solid lines are the model under RC setting while the dotted lines are the model whose eigenvalues are allowed to learn.

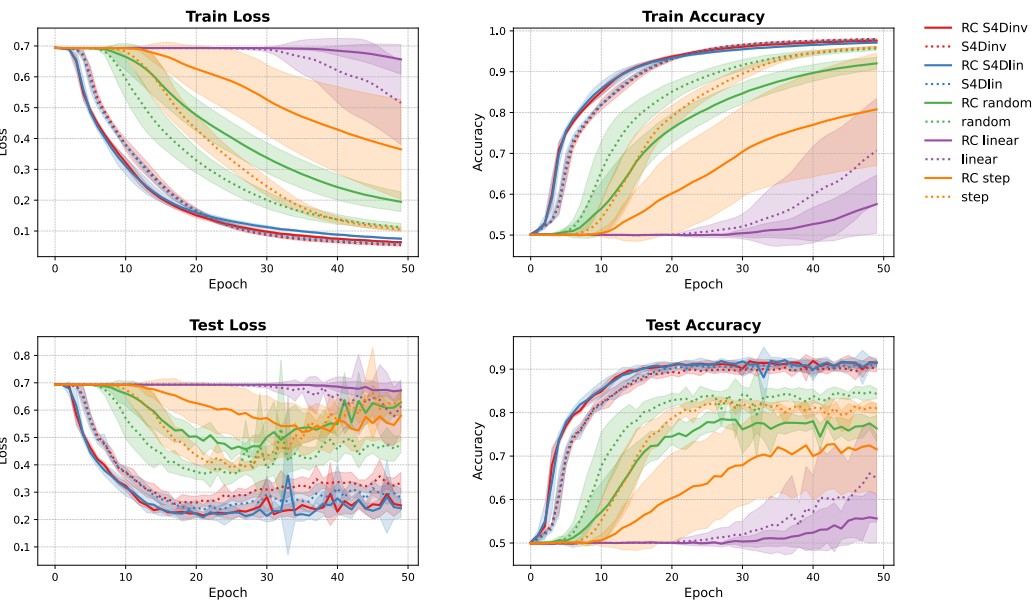

Figure 15: Test loss and accuracy of during the learning of PathX task. The upper (lower) row indicates the metrics of train (test). The column corresponds to each metric. The horizontal axes are epochs. The colors indicate eigenvalue realizations. The solid lines are the model under RC setting while the dotted lines are the model whose eigenvalues are allowed to learn.

# F   FURTHER DISCUSSION

## F.1   RELATED RESEARCHES

Related research topics include the vanishing and exploding gradient problems (Zucchet & Orvieto, 2024; Park et al., 2023; Sokół et al., 2019). These studies focused on multilayer networks, where gradients accumulate multiplicatively as errors propagate through the layers. For RNNs in particular, the problem is tied to the traditional backpropagation through time (BPTT) algorithm, which treats the number of recurrent steps as the number of gradient accumulations. In the model examined in this study, the notion of "multiple layers" can be understood in the two ways described above. One interpretation (a) is to deepen the model by stacking several distinct SSMs along the depth direction. In this view, the number of layers corresponds to the number of different SSM modules. The other interpretation (b) focuses on the temporal evolution within a single SSM. In this case, the number of layers is regarded as the number of time-propagation steps. For both interpretations, we explain how the problems of vanishing and exploding gradients arise by using the results obtained in this study. In order to do that, we first revisit the gradient expressions analyzed in this work. For a target time series $\mathbf{y}$, the gradient that an SSM layer sends to the previous layer is given by expression

$$G(\boldsymbol{y}) \quad = \quad \frac{2}{T} \boldsymbol{X}^\top \boldsymbol{e}, \tag{52}$$

where $\boldsymbol{e} = (\boldsymbol{X}\boldsymbol{\theta} - \boldsymbol{y})$ and $\mathbf{e}$ denotes the error time series. By simplifying this expression analytically, we obtain expression

$$G(\boldsymbol{y}) \quad = \quad 2\langle u^2 \rangle B \boldsymbol{V}^\top \boldsymbol{e}, \tag{53}$$

where $\boldsymbol{e} = (B\boldsymbol{V}\boldsymbol{\theta} - \boldsymbol{V}(\boldsymbol{V}^\top\boldsymbol{V})^{-1}\boldsymbol{V}^\top\boldsymbol{\alpha})$ and $\boldsymbol{V} = \begin{pmatrix} \lambda_1^{T-1} & \lambda_1^{T-2} & \cdots & \lambda_1^{1} & 1 \\ \lambda_2^{T-1} & \lambda_2^{T-2} & \cdots & \lambda_2^{1} & 1 \\ \vdots & & & & \\ \lambda_N^{T-1} & \lambda_N^{T-2} & \cdots & \lambda_N^{1} & 1 \end{pmatrix}^\top$ . Based on these formulas, we discuss how the error information decays.

We begin with case (b). Here, the Vandermonde matrix $\mathbf{V}$ characterizes how the error information decays through BPTT. The mechanisms that lead to the loss of error information can be classified into two types. The first is due to numerical precision, where values fall below precision of computers and are truncated. The second is independent of numerical precision and arises from the structure of the model itself. The mechanism caused by numerical precision can be explained by examining a single row of $\mathbf{V}^\top$. During BPTT, the exponent applied to each eigenvalue increases with every propagation step. When the absolute values of the eigenvalues are raised to these powers, they can fall below the numerical precision, causing the loss of information. The rate of decay can be described as following an exponential law. In contrast, the mechanism caused by the model structure does not necessarily follow exponential decay. As shown in Section 3.2, the decay of this matrix can be evaluated by the MF. This fact can also be confirmed directly from the formulas. According to the BPTT mechanism, the state time series generated by injecting the error sequence into the SSM as inputs corresponds to the gradient for the SSM layer. This resulting state sequence is precisely $B\boldsymbol{V}^\top \boldsymbol{e}$ in Eq. 53. Here we consider computing gradients utilizing the full time series of state values. By evaluating how accurately the error information can be reconstructed from the state time series, we can assess how much the error information decays through BPTT. Since this is exactly the definition of the MF, the decay of error information can be considered captured by MF.

Earlier studies on RNNs often treated the mechanism caused by the model structure in the same way as the problem of numerical precision by assuming the decay rate as an exponential. However, from the perspective of MF, such an assumption is not always correct. For example, when the delay is small, the MF remains close to 1, as shown in Fig. 1. Accordingly, using MF may enable a more accurate evaluation of how error information decays.

Gradient explosion can be analyzed in a way similar to the gradient vanishing. The mechanism caused by the numerical precision similarly induces the gradient explosion because of the exponential growth of values during propagation for each eigenvalue. However, from the MF perspective, the mechanism caused by the model structure does not induce the gradient explosion, because MF normalizes the state values, evaluating only the decay pattern regardless of the actual magnitude of the states.

Next, we consider the case of (a). Here, vanishing and exploding gradients arise when gradients from multiple SSM layers are multiplied together. This reduces to the classical problem of vanishing and exploding gradients in deep networks.

## F.2 SCOPE OF THIS STUDY

The focus of this study is not on the problems of gradient vanishing or explosion described above. Rather, it is on identifying the factors that determine whether gradient-based optimization methods can successfully learn. To this end, we focus on gradient computations performed for each input time series. However, our analysis does not consider only the vanishing of error information during gradient computation. The most significant difference from prior approaches is that we do not restrict the analysis to a single gradient update. Instead, we treat the influence on subsequent gradient updates in a unified manner. This allows us, for example, to determine whether a gradient-based optimization method can recover from vanishing gradients in the next update step. To address this problem, we reinterpret the error information not as it is, but as a function of the input, clarifying how the input time series are processed during backpropagation. This clarifies how input information drives gradient updates and how subsequent inputs are processed during later gradient computations conditioned on the current update. Consequently, it reveals the overall learning mechanism in a chain-like manner. Such an approach has not been adopted in prior work on gradient vanishing, as those methods do not unfold the error information with respect to the input.

In this study, we take a further step beyond the analysis of gradient vanishing in case (b). By expanding $e$ and expressing the target time series in terms of the input time series, we make this analysis possible. Rather than treating the error information and input independently, we represent it solely in terms of the input, simplifying the explanation. This explicitly shows that normalized $\mathbf{V}$, including MF, appears in the gradient computation. We prove that the gradient direction is determined by the product of the matrix $\boldsymbol{V}(\boldsymbol{V}^\top \boldsymbol{V})^{-1}\boldsymbol{V}^\top$, capturing the decay of MF, and the series $\boldsymbol{\alpha}$, quantifying the amount of input information contained in the target time series. This clarifies how input information is utilized in gradient computation within SSM layers.

Using the results of this analysis, we demonstrate that the initial values play a crucial role in the success of learning. Previous studies suggest that appropriate initialization is important to avoid gradient vanishing, but this does not imply that improper initialization necessarily leads to learning failure. If a mechanism existed that allows backpropagation to overcome vanishing gradients and prevents them from occurring during learning, then gradient vanishing at the initial stage would not be problematic. However, our results show that, in SSMs, no such mechanism exists when using gradient-based optimization methods. This result suggests the learning mechanism of SSMs and the importance of initialization for a successful learning.

## F.3 NUMERICAL EXPERIMENTS

In this study, we adopted the hyperparameters used in prior work. We keep the configuration consistent across tasks and no model-specific tuning is performed, changing only the random seed. We consider the hyperparameter settings presented in the prior work to be close to optimal, and therefore believe that further performance improvements are extremely difficult under the conventional full-parameter training framework. However, our results for S4D variants sometimes differ from those reported in the S4D paper (Gu et al., 2022a). The issue of being unable to reproduce the results for S4 has also been posted as a question on the GitHub repository. It has been pointed out that such irreproducibility may stem from differences in hardware or software, such as GPU, PyTorch, or CPU versions. In our environment, the CPU differs from that used in the original implementation. To verify this, we ran multiple trials with different random seeds; however, the results remained irreproducible for some tasks. We consider such discrepancies to be within a reasonable range, consistent with discussions on GitHub. While it is possible that parameter re-tuning would be necessary in a different environment, we did not pursue this, as it falls outside the main focus of our study.

## F.4 APPLICABILITY TO BROADER MODELS

Our study based on MF has the potential to be applied to broader models. In particular, since the attention mechanism in Transformers can be regarded as a linear RNN, the results derived in this work immediately enable a direct comparison between Transformers and SSMs. An SSM possesses $N$ nodes with distinct eigenvalues, so the theoretical upper bound of its MC is $N$. In contrast, the attention mechanism corresponds to a linear RNN with $N = 1$, where the single eigenvalue is equal to 1. Therefore, in order for a Transformer to achieve an MC comparable to that of an SSM, it would require $N$ independent attentions. However, if all the eigenvalues of these attention are identical, the Vandermonde matrix which is equivalent to the states time series suffers a rank deficiency and its rank collapses to 1. Since the rank of the Vandermonde matrix determines the upper bound of MC in the analytical solution, increasing the number of attentions does not improve MC. This could be one of the reasons why SSMs can surpass Transformers.

Moreover, the indicator MF can potentially be applied to other SSMs such as Mamba, Mamba2, and Gated Delta Networks (Gu & Dao, 2023; Dao & Gu, 2024; Yang et al., 2025) . These models incorporate nonlinear operations into the RNN state updates and can therefore be regarded as nonlinear RNNs. For this reason, the results established in this study cannot be directly extended to them. Frameworks like IPC, which can accommodate nonlinearity, may offer a path forward. In S4 and Mamba, the number of nodes $N$ takes a relatively large value (e.g., $N = 32$), whereas in Mamba2, which is based on the Transformer architecture, we have $N = 1$. Since both the sum of IPC and MC have the same upper bound, the state size $N$, the significant drop in the theoretical capacity of Mamba2 is noteworthy. This observation indicates that the high performance of Mamba2 is likely supported by the nonlinearity or other architectural elements. In addition, the total effective IPC can numerically differ from the theoretical upper bound. This numerical upper bound can be regarded as the number of effective nodes. Making the RNN nonlinear may increase the number of effective nodes. Additionally, nonlinear operations may also change the components of linear information processing. These effects together suggest that a more diverse information processing may be realized in nonlinear SSMs. Therefore, long range dependencies may deteriorate while the scores on natural language processing are enhanced (Yang et al., 2025).

