# OpenReview forum: "Memory Determines Learning Direction: A Theory of Gradient-Based Optimization in State Space Models"
_ICLR.cc/2026/Conference — Submitted to ICLR 2026_

### Official Review · Reviewer_NU5d · 2025-10-21

**Soundness:** 2
**Presentation:** 2
**Contribution:** 2
**Rating:** 4
**Confidence:** 3

**Summary:**

This paper develops a theoretical framework explaining how memory determines the learning dynamics of state space models. By analyzing how input sequences are encoded in model states, the authors show that the memory function can quantify a model’s capacity to store and utilize past information. They prove the theoretical equivalence between the structured state space sequence model and its diagonalized variant and demonstrate that the initial eigenvalue configuration—which governs memory length—is crucial for successful training. Their analysis reveals that longer initial memory, even at the cost of accuracy, leads to more effective gradient-based learning. Building on this insight, they propose a training strategy with fixed recurrent weights, which preserves the optimal memory structure, reduces overfitting, and accelerates convergence. Experiments on the Long Range Arena benchmark validate the theory and highlight the practical benefits of memory-aware initialization and training.

**Strengths:**

This paper makes a clear and original theoretical contribution to understanding the learning dynamics of state space models. The authors rigorously define memory capacity through the lens of the memory function and use it to derive analytical insights into gradient-based optimization—a direction that has not been formally established in prior work. The study is conceptually elegant and technically solid, offering a mathematically grounded explanation of why initialization plays a decisive role in successful training.

It is particularly interesting that the paper reemphasizes the importance of initialization, echoing but also extending the intuition behind HiPPO by connecting it explicitly to gradient flow and memory preservation. The proofs are logically consistent and supported by well-designed empirical validation on long-sequence benchmarks. The exposition is clear and well-organized, making complex ideas accessible.

**Weaknesses:**

While the theoretical framework is rigorous and illuminating for linear or non-gated state space models, it does not generalize easily to gated architectures such as Mamba or Gated DeltaNet. In these newer models, the system dynamics—and therefore the eigenvalues—depend on the input, breaking the assumption of fixed linear recurrence that underlies the present analysis. Extending the theory to handle input-dependent dynamics remains a key open challenge. Developing a suitable mathematical framework to describe and analyze such cases would be essential for broadening the applicability of the paper’s results to the full family of modern state-space architectures.

**Questions:**

It is important to discuss the performance gap between S4 and mamba / GDN (because gated delta net has been adopted in large scale open source models.) There should be some discussion why GDN is better than S4.

---

> ### Author Response · Authors · 2025-11-18
> **Response to Reviewer NU5d**
>
> We sincerely appreciate the time and effort you have devoted to reviewing our manuscript.
> We are also grateful for your summary of our work and for highlighting its main strengths.
> We have addressed the concern and question you raised, and we hope that our responses provide clear explanations.
> We would be grateful for your continued engagement during the discussion phase and welcome any additional feedback you may have.
>
>
> >While the theoretical framework is rigorous and illuminating for linear or non-gated state space models, it does not generalize easily to gated architectures such as Mamba or Gated DeltaNet. In these newer models, the system dynamics—and therefore the eigenvalues—depend on the input, breaking the assumption of fixed linear recurrence that underlies the present analysis. Extending the theory to handle input-dependent dynamics remains a key open challenge. Developing a suitable mathematical framework to describe and analyze such cases would be essential for broadening the applicability of the paper’s results to the full family of modern state-space architectures.
>
> In fact, our theoretical results based on MF could potentially be directly applied to nonlinear SSMs such as Mamba or Gated DeltaNet.
> These SSMs can be interpreted as nonlinear RNNs.
> Because the MF can be assessed similarly, the insights on initialization and the importance on memory length may remain valid.
> However, analyzing nonlinear RNNs not only requires focusing on MF but also the memory arising from the nonlinear information processing.
> Unlike in linear RNNs, it remains unclear which factors are critical for achieving high performance in nonlinear memory.
> Identifying what aspects of memory are essential for nonlinear RNNs is an ongoing research topic.
> For this reason, we restrict our analysis to linear RNNs in the present study.
> The measure for investigating nonlinear information processing already exist and are referred to as
> information processing capacity (IPC) that is an nonlinear extension of MF.
> It is potentially possible to measure the IPC of those nonlinear SSMs.
>
>
> >Questions:
> It is important to discuss the performance gap between S4 and mamba / GDN (because gated delta net has been adopted in large scale open source models.) There should be some discussion why GDN is better than S4.
>
> Thank you for the question. The reasons why models such as Mamba and GDN achieve stronger performance are, in the context of our study, consistent with the understanding presented in previous works.
> According to prior research, one limitation of S4 is that its state updates are input-independent, which can reduce flexibility when tasks require behavior that depends on the current context. Mamba and GDN address this limitation by incorporating input-dependent gating mechanisms, which allow for more dynamic selection or suppression of information. Adding this gating mechanism does not remove the main strengths of S4. However, it allows Mamba and GDN to perform better on tasks that need flexible behavior depending on the context.
>
> In the context of our study, Mamba and GDN can be collectively evaluated as nonlinear RNNs under the IPC metric.
> Both models may share specific nonlinear information processing mechanisms that contribute to their performance improvements in addition to the linear information processing that S4 already has.
> From the perspective of IPC and MF, the upper bound of IPC is determined by the number of nodes.
> Therefore, if S4 and Mamba have the same number of nodes in their SSMs, their theoretical maximum IPC is also the same.
> However, in practice, the total effective IPC can differ from this theoretical upper bound. This total can be regarded as the number of effective nodes. Making the RNN nonlinear may increase the number of effective nodes. Additionally, nonlinear operations may also enhance the diversity of linear information processing. These effects together suggest that a more desirable information processing may be realized in practice.

---

### Official Review · Reviewer_WKGg · 2025-10-27

**Soundness:** 3
**Presentation:** 2
**Contribution:** 1
**Rating:** 2
**Confidence:** 4

**Summary:**

Deep state-space models have received significant interest recently due to their empirical performance. However, their learning dynamics are still poorly understood theoretically. This paper aims to bridge this gap by investigating them from the perspective of the memory function. This approach enables highlighting the importance of initialization for successful learning of these architectures. The theoretical results are complemented with empirical analysis of the memory function for different types of initializations, as well as results emphasizing the benefits of freezing the SSM eigenvalues in terms of learning stability and thus learning performance.

**Strengths:**

- Learning dynamics of SSMs are a complicated process; gaining understanding of them is an important direction that this paper pursues.
- The memory function perspective the paper leverages is in principle an interesting lens through which to look at SSMs.
- The empirical results showing the stability benefits of RC on SSMs are interesting.

**Weaknesses:**

The theory introduced in the paper reads to me as sophisticated mathematics to arrive at conclusions that could be simply derived from the vanishing gradient perspective: initialization is indeed critical in RNNs to ensure that correlations between distant time steps can be learned. It is unclear to me that the theory adds anything beyond that.

To my understanding, this may occur because previous results from the literature are misrepresented. While I found many inaccuracies in citations of previous work, I'll highlight the most important ones below:
- **The problem the authors study is the vanishing gradient problem**. In lines 461-462, the authors write "Vanishing gradients may not occur because the only case in which G(y) = 0 […]". Vanishing gradients are not about the total gradient being close to 0, but about error signals transmitted over τ time steps decreasing exponentially with τ.
- **The benefits of diagonality are theoretically relatively well understood**. In line 50, it is written that "theoretical explanation justifying [diagonality] is still lacking". This is not accurate. The S4D and LRU papers argue about the (almost sure) equivalence of these classes in terms of representation. Orvieto et al. 2024 shows the benefits of complex-valued over real-valued diagonal recurrence in terms of reachability by gradient descent. Zucchet & Orvieto 2024 shows the optimization benefits of the normalization/reparametrization that diagonality enables. While precise understanding of the learning dynamics remains poorly understood, the existing understanding already enables explaining most (if not all) of the theoretical conclusions of the paper.

**Questions:**

The results reported in Table 1 for S4D variants sometimes differ significantly from those reported in the S4D paper, bridging the gap with the RC results. Can the authors elaborate on that?

In line 312 it is mentioned that only the final weights are learned in the RC experiments, whereas line 465 says that 10% of the parameters are not learned with RC (which suggests only freezing eigenvalues and learning the rest of the network). Which one is correct?

Could the authors elaborate on why equation 5 holds? The left term is an infinite sum of terms that are smaller than 1; why is that smaller than $N$?

---

> ### Author Response · Authors · 2025-11-18
> **Response to Reviewer WKGg (1/3)**
>
> We sincerely thank you for taking the time to review our manuscript.
> We also appreciate your thoughtful summary of the paper and your clear identification of its strengths.
> We have carefully addressed the questions and concerns you raised, and we hope that our responses sufficiently clarify them.
> We would greatly appreciate your participation in the discussion phase as well, and we welcome any further comments you may have.
>
>
> >The problem the authors study is the vanishing gradient problem. In lines 461-462, the authors write "Vanishing gradients may not occur because the only case in which G(y) = 0 […]". Vanishing gradients are not about the total gradient being close to 0, but about error signals transmitted over τ time steps decreasing exponentially with $\tau$.
>
> Thank you for the comment.
> In conventional settings, the problem arises in feedforward networks through the propagation of error signals across multiple layers, and in RNNs through backpropagation through time over $\tau$ steps. This can result in numerically extremely large or small values that exceed the precision of the computing device, leading to failure. Our study, however, addresses a different issue that occurs prior to these numerical problems. For example, the traditional vanishing gradient problem corresponds to powers of eigenvalues; as these powers numerically approach zero, they can fall below machine precision, potentially failing to contribute to information processing capabilities. In contrast, the problem we highlight is independent of numerical precision. It does not arise because a specific eigenvalue's power becomes zero, but because of a design flaw in the memory architecture when considering the single entire SSM layer.
> Consequently, during gradient computation, label information may not be preserved. This can occur even if numerical vanishing or exploding gradients are avoided.
> We have theoretically shown that sufficiently long memory architecture is required to prevent this issue. Therefore, this is not the conventional vanishing gradient problem, and we have kept this discussion to a minimum.
>
> However, in the sense of failure in propagating label information through gradients, there is a conceptual relation to conventional problems. Accordingly, in the discussion paragraph starting at line 450, we use our theoretical results to discuss traditional vanishing and exploding gradient problems.
> In this paragraph, since the problem we proved arises when considering the entire SSM layer, we attempt to explain conventional issues from the same perspective using the mechanism discovered in this study. Therefore, unlike the traditional view, we do not focus on error propagation between individual parameters.
> Accordingly, strictly speaking, this is a different problem from the conventional vanishing gradient.
> Under this new perspective, we consider situations analogous to the vanishing of gradients, i.e., when the total gradient $G(y) = 0$. The key difference is that, unlike the conventional case, such a situation does not occur before training is completed.
> Nonetheless, there may still be failure in propagating label information as error signals.
> This failure can only be concluded by considering the SSM layer as a whole.
>
> To compare with the problem discussed in this paper, we considered the vanishing gradient problem at the level of a single SSM layer.
> As this is not the main point of our paper, we did not explicitly extend the conventional definition of vanishing gradients.
> We recognize that this may have caused confusion. In the revised manuscript, we will clearly define this extended notion of vanishing gradients, compare it in detail with the problem highlighted in our study, and discuss the differences between our problem and the conventional problem. Furthermore, we will include a discussion on how the insights obtained in this study can also inform understanding of the conventional problem.

---

> > ### Author Response · Authors · 2025-11-18
> > **Response to Reviewer WKGg (2/3)**
> >
> > >The benefits of diagonality are theoretically relatively well understood. In line 50, it is written that "theoretical explanation justifying [diagonality] is still lacking". This is not accurate. The S4D and LRU papers argue about the (almost sure) equivalence of these classes in terms of representation.
> >
> > Thank you for the comment.
> > In the study of S4D, the focus is placed on the HiPPO matrix and its specific properties.
> > Therefore, the method cannot be extended to general SSMs with arbitrary weights, which limits its universality and significantly restricts its applicability.
> > For example, at the end of our manuscript, we discuss the possibility of achieving further memory extension.
> > If one remains committed to the HiPPO matrix, such improvements cannot be discussed since it prevents the use of potentially better weight parameterizations.
> > Moreover, S4D argues that a Normal HiPPO matrix can be replaced with a matrix having the same eigenvalues because they are unitarily equivalent.
> > However, this is fundamentally different from asserting strict equivalence in terms of how accurately past input information can be preserved in the states and reproduced from the current states,
> > as discussed in our work.
> > The claim of LRU paper also focuses on computational acceleration or empirically performance
> > and does not indicate any theoretically equivalence in terms of information processing as measured by MF.
> > Therefore, the results of these researches cannot be applied to subsequent gradient evaluation in our results.
> >
> >
> > >Orvieto et al. 2024 shows the benefits of complex-valued over real-valued diagonal recurrence in terms of reachability by gradient descent. Zucchet & Orvieto 2024 shows the optimization benefits of the normalization/reparametrization that diagonality enables.
> >
> > Thank you for the information. We also acknowledge that appropriately designing the eigenvalues of an SSM is important, as achieving longer memory structures requires exploring more suitable eigenvalues.
> > This necessity is grounded in our proof in the main text that long memory structures are required, and the demonstration that even longer memory structures can be realized through training.
> > However, our study does not aim to discuss the benefits of complex-valued over real-valued diagonal recurrence, nor the optimization advantages afforded by the normalization/reparametrization that diagonality enables.
> > Rather, our focus on S4D is motivated by a different consideration.
> > We believe that the theoretical equivalence between S4D and S4 from the perspective of information processing is significant in itself.
> > In addition, by demonstrating this equivalence, we are able to identify fundamental conditions required for the memory that are essential to the successful training of both S4 and S4D.
> > Furthermore, from the perspective of MF, the need for eigenvalue design to achieve longer memory structures has not been addressed in prior work. Therefore, we believe that our contribution does not overlap with the content of the studies you referenced.
> >
> >
> > >While precise understanding of the learning dynamics remains poorly understood, the existing understanding already enables explaining most (if not all) of the theoretical conclusions of the paper.
> >
> > It is clear that the relevance of initial memory architecture to the task performance has been empirically suggested by studies such as S4 and S4D.
> > However, to the best of our knowledge, no prior work has theoretically explained the
> > reason why the choice of initial parameters is critically important for successful training.
> > Our results clarify this by showing that any gradient based learning has no explicit mechanism to reflect the label information associated with inputs that the SSM fails to memorize in forward propagation.
> > Moreover, we are not aware of any literature that empirically suggests that learning eigenvalues may not be necessary with multiple advantages based on the learning mechanism discussed in our study.
> > If you are aware of any relevant work that should be taken into account in our discussion, we would be grateful for your guidance.
> > If there are specific points in the papers you mentioned that should be discussed in detail, we would greatly appreciate it if you could point them out, as this would help us deepen our understanding.

---

> > > ### Author Response · Authors · 2025-11-18
> > > **Response to Reviewer WKGg (3/3)**
> > >
> > > Questions:
> > > >The results reported in Table 1 for S4D variants sometimes differ significantly from those reported in the S4D paper, bridging the gap with the RC results. Can the authors elaborate on that?
> > >
> > > Thank you for the comment. As a preliminary step, we reran the experiments using almost the same program environment and hyperparameters as recommended on GitHub codes (https://github.com/state-spaces/s4) provided by the authors of original S4 and S4D papers.
> > > The issue of being unable to reproduce the results for S4 has also been posted as a question on the GitHub repository.
> > > It has been pointed out that such irreproducibility may stem from differences in hardware or software, such as GPU, PyTorch, or CPU versions.
> > > In our environment, the CPU differs from that used in the original implementation.
> > > To verify this, we ran multiple trials with different random seeds; however, the results remained irreproducible for some tasks.
> > > We consider such discrepancies to be within a reasonable range, consistent with discussions on GitHub.
> > > While it is possible that parameter re-tuning would be necessary in a different environment, we did not pursue this, as it falls outside the main focus of our study.
> > >
> > >
> > >
> > > >In line 312 it is mentioned that only the final weights are learned in the RC experiments, whereas line 465 says that 10% of the parameters are not learned with RC (which suggests only freezing eigenvalues and learning the rest of the network). Which one is correct?
> > >
> > > Thank you for the question. We would like to clarify that the term output weights does not refer to the final layer of the network. Rather, it refers to *the output layer of each individual SSM*.
> > > In the RC experiments, only these output weights within each SSM are learned, while the eigenvalues remain fixed.
> > > We hope this clarifies the apparent discrepancy between lines 312 and 465.
> > >
> > > >Could the authors elaborate on why equation 5 holds? The left term is an infinite sum of terms that are smaller than 1; why is that smaller than ?
> > >
> > > Thank you for the question.
> > > The validity of Equation (5) follows from established results in the literature [1].
> > > The relevant proof can be found on page 16, Equation (23), of the cited work.
> > >
> > > [1] Jaeger, Herbert. "Short term memory in echo state networks." (2001).

---

> > > > ### Comment · Reviewer_WKGg · 2025-11-18
> > > >
> > > > I thank the authors for their detailed rebuttal.
> > > >
> > > > Unfortunately, I am still not convinced that the theoretical insights are significantly different from the ones the vanishing gradient perspective gives. In my opinion, this is possible as the authors consider a restricted version of the vanishing gradient problem is: vanishing gradients are not only a "numerical problem". Even with infinite precision, label information would barely flow to far away time steps and it is practically impossible to learn long-range dependencies (which is the same conclusion the authors arrive to using the MF framework).
> > > >
> > > > To me, this is a fundamental issue that requires important rewriting of the story and the claims of the paper. I therefore keep my score as it is.

---

> > > > > ### Author Response · Authors · 2025-11-18
> > > > >
> > > > > Thank you for your feedback.
> > > > > If possible, could you also address the following point? We believe that the passage below is not related to the definition of the vanishing gradient problem.
> > > > >
> > > > > >However, to the best of our knowledge, no prior work has theoretically explained the reason why the choice of initial parameters is critically important for successful training. Our results clarify this by showing that any gradient based learning has no explicit mechanism to reflect the label information associated with inputs that the SSM fails to memorize in forward propagation. Moreover, we are not aware of any literature that empirically suggests that learning eigenvalues may not be necessary with multiple advantages based on the learning mechanism discussed in our study. If you are aware of any relevant work that should be taken into account in our discussion, we would be grateful for your guidance.

---

> > > > > > ### Comment · Reviewer_WKGg · 2025-11-18
> > > > > >
> > > > > > > However, to the best of our knowledge, no prior work has theoretically explained the reason why the choice of initial parameters is critically important for successful training.
> > > > > >
> > > > > > Given the vanishing (and exploding) gradient problem, some papers have investigated how to best initialize recurrent neural networks. The idea is to initialize the recurrence so that the eigenvalues of the recurrent Jacobian are close to 1, e.g. [1](https://proceedings.mlr.press/v70/vorontsov17a.html) [2](https://proceedings.mlr.press/v48/henaff16.html) [3](https://proceedings.mlr.press/v80/helfrich18a.html) [4](https://proceedings.mlr.press/v48/arjovsky16.html). Regarding SSMs, the fact that eigenvalues need to be close to the unit circle at initialization to avoid vanishing gradients is for example discussed in the LRU paper [5](https://arxiv.org/abs/2303.06349) (Section 3.3).
> > > > > >
> > > > > > > We are not aware of any literature that empirically suggests that learning eigenvalues may not be necessary with multiple advantages based on the learning mechanism discussed in our study.
> > > > > >
> > > > > > To the best of my knowledge this was not reported before. However, I don't see how the theory the author proposed can explain this. The most related finding that can explain this is [6](https://arxiv.org/abs/2405.21064). They found the learning of the angle of complex eigenvalues to be prone to instability, particularly when eigenvalues are similar to each other; fixing them might avoid these instabilities.

---

> ### Author Response · Authors · 2025-11-21
>
> Thank you for your response and valuable feedback.
> Based on the prior research you have presented, we would like to restate the novelty and importance of our study.
>
> # Importance of MF
>
> First, we have sought to clarify how error information decays during backpropagation (BP), and we have identified that this decay pattern is actually represented by the MF.
> This means that evaluating MF may help us for elucidating the learning mechanisms based on gradient computation,
> providing two important insights for gradient computation.
> The first one comes from the definition of MF, which represents how accurately one can reconstruct past inputs from the current state, as stated in both our manuscript and our initial response.
> The second one is that MF indicates the minimum level of memory that a SSM can hold.
> This is because that MF usually assumes i.i.d. inputs, while real inputs often have correlations. Correlated inputs have longer memory than i.i.d. inputs due to their correlation, so their memory range goes beyond what MF measures.
> Even with this minimum level of MF, we can observe that reconstruction is nearly perfect for small delays as can be confirmed in Figure 1 of our study and figures in prior research [1,2].
> These researches have numerically demonstrated that the length where MF stays near 1 is roughly proportional to the number of nodes.
> This suggests that SSMs have the potential to achieve at least this level of performance.
>
> # Why initial parameters are important
>
> Next, from the gradient expression we derived, we proved that BP has no explicit mechanism to extend the memory structure.
> We explained this by proving that MF plays a role in selecting label information.
> We consider that this is the reason why initial parameters are important and the RC setting can be applied.
> In our paper, we have consistently focused on this problem, and based on the mechanism of this problem, we have conducted numerical experiments and performed verification.
>
> # Implications for vanishing gradients
>
> In addition, the selection by the MF can also provide new explanation for the vanishing gradient problems.
> Specifically, the selection by the MF means that MF truncates the label information linked to inputs that the SSM fails to memorize in forward propagation.
> Using this fact, we could reveal some important characteristics about the vanishing gradient problem.
> For example, during BP, the error information  for inputs with small delays can be perfectly reflected.
> As a result, this allows us to discuss the vanishing gradient problem from the perspective of MF and may allow us to explain what kind of information is discarded more in detail.
>
> # What we focus here
>
> However, the main focus of our research is not to address the vanishing gradient problem.
> We want to consider, when the error information is lost, whether BP has the ability to recover from the failure.
> Our results show that BP can solve this problem only in limited cases.
> Using the MF, we could prove that if a memory about a certain input exist initially,
> the error information regarding the input may be improved by increasing the memory accuracy.
> But if the memory of the input does not exist initilly,
> the error infomation cannot be explicitly imporoved at all.
> This arises from the fact that BP has no explicit mechanism to extend the memory structure.
> Because this is the problem of BP, we argue that our study could reveal a part of learning mechanim.
> We consider that if we rely solely on the idea that error information vanishes,
> it may not be possible to explain this issue clearly from the problem statement through to the conclusion suggesting the learning mechanism.
>
>
> # Acknowledgement
>
> Finally, we greatly appreciate your comments on the vanishing gradient problem.
> For example, we believe that the research you have introduced provides valuable insights into how to design the initial value from the perspective of the vanishing gradient problem.
> Since our problem also highlights the importance of initial values, we believe that we should certainly refer to approaches for addressing the vanishing gradient problem when selecting better initial values from our perspective.
>
>
> > To the best of my knowledge this was not reported before. However, I don't see how the theory the author proposed can explain this.
>
> We suggest that learning eigenvalues may not be necessary with multiple advantages by using the fact that BP has no explicit mechanism to extend the memory structure.
> Based on this theoretical results, we predicted that RC setting might be beneficial.
> This prediction is discussed in the sections 3.2 "Training eigenvalues" and 3.3.
> Using the numerical experiments, we actually demonstrated that MF changes little and the prediction based on the theory was correct.
>
>
> [1] Jaeger, Herbert. "Short term memory in echo state networks." (2001).
>
> [2] Guan, Jing Chuan , et al. "How noise affects memory in linear recurrent networks." #i{PHYSICAL REVIEW RESEARCH} 7.2(2025):023049.

---

### Official Review · Reviewer_kJk2 · 2025-10-28

**Soundness:** 2
**Presentation:** 1
**Contribution:** 2
**Rating:** 4
**Confidence:** 3

**Summary:**

This paper presents a theoretical analysis of gradient-based learning in State Space Models (SSMs), using the memory function (MF) and memory capacity (MC) concepts originally developed in reservoir computing (RC).

The authors first restate that the dense S4 SSM layer is theoretically equivalent to its diagonalized version (S4D). They then analyze the MF of diagonal S4 variants and argue that these architectures achieve better reconstruction of delayed inputs than naïvely designed SSMs. They further relate the gradient of the loss with respect to the output parameters to the MF, both in the linear and nonlinear output cases, showing that learning dynamics depend on the initial MF profile. Based on these analyses and empirical observations, they argue that training the recurrent weights is unnecessary and potentially harmful.

Empirical evaluations on the Long Range Arena (LRA) benchmark confirm the theoretical claims: freezing the recurrence (RC-style training) improves convergence speed and generalization, and structured diagonal SSMs (S4Dinv, S4Dlin) outperform simpler baselines.

**Strengths:**

- Establishes an interesting conceptual link between state-space models and the memory function analyses used in reservoir computing.

- The empirical computation of MF for S4D variants is novel and clearly shows their advantage in preserving long-term dependencies.

- The experiments show evidence that freezing the recurrence can improve stability and generalization.

**Weaknesses:**

- **Lack of clarity in the theoretical part.** The theoretical statements are all expressed in dense natural language, making it hard to isolate the main claims and their assumptions. There is a lack of formal structure (e.g., lemmas or theorems) that would help organize the theoretical contributions. In particular, $\alpha_\tau$, a key quantity in the analysis, is never formally defined. It is unclear whether it corresponds to the classical convolution kernel of the SSM or to another object. It is introduced (L172) after a citation to *Dambre et al.* (2012), but the precise result being referenced from that work is not specified.

  - **S4D is equivalent to S4.** The assumptions underlying this claim are not discussed. It seems to rely on the uniform i.i.d. input assumption, which should be made explicit and justified. Moreover, according to the empirical results reported in the paper (and in previous literature on S4 and its diagonal variants), S4 outperforms S4D. It would therefore be important to clarify under which conditions this theoretical equivalence is supposed to hold. The authors also state that “since the MF $M[u_{t-\tau}]$ is statistically equivalent to the norm of $\alpha_\tau$” (L176) without citing where this result originates from.

  - **Initialization for successful learning.** The derived gradient formula (Eq. 12) is not clearly connected to the final claim that one should prioritize MF profiles with longer delays (even if their amplitudes are small). If this is a central claim, it could be empirically tested by comparing systems along the Pareto front of MF profiles (accuracy vs. long-delay memory) to verify whether emphasizing long-delay memory indeed improves learning.

- **Fragile link between theory and experiments.** The main empirical claim—that freezing the recurrence improves performance—is only loosely justified by the theory. The explanation of an “unintended extension of the MF” (L302) would be stronger if formalized or tested empirically. If the underlying mechanism is instead regularization or overfitting mitigation, the authors should compare against standard techniques (weight decay, dropout, etc.).

- **Reservoir computing motivation.** The appeal to reservoir computing (RC) as motivation for not training the recurrence is weak. In RC, the motivation is computational efficiency (avoiding backpropagation through time), not improved accuracy. Here, since gradients are still computed through the recurrence, this rationale is only partly convincing. A more convincing motivation would be to analyze how to identify and remedy the pathologies that arise when training the recurrence.


Minor issues and typos:
- L086: “It **builts** on continuous-time linear SSMs” → *builds*
- L117: “The emulation is conducted” → not clear what *emulation* means here
- L174: “using a **constatnt** sequence” → *constant*
- L199: “index **indentifying** a certain SSM” → *identifying*
- L214: "here T is the output dimension of SSM" → based on the context, it seems that T is the sequence length, not the output dimension
- L427: “**suggestting** that learning eigenvalues…” → *suggesting*
- L466: “**Altough** the reduction is small” → *Although*
- L632: “uniform **randomo** distribution” → *random*
- L674: “one SSM **lyaer**” → *layer*
- L683: “numerical **unstability**” → *instability*
- L698: “structured eigenvalues **indicates** significantly more…” → plural agreement → *indicate*

**Questions:**

1. **Scope of gradient analysis**
   The theoretical analysis focuses on gradients with respect to the output-layer parameters, even though the models are trained with full backpropagation through the recurrence. Why not also analyze gradients with respect to the **input** weights, which could reveal how input encoding interacts with the memory structure? Do you expect such an analysis to yield additional insights?

2. **Effect of regularization tuning**
   Did you test whether the reported improvements with frozen recurrent weights persist when regularization hyperparameters (e.g., weight decay, dropout) are more carefully tuned in the fully trainable setting?

---

> ### Author Response · Authors · 2025-11-18
> **Response to Reviewer kJk2 (1/3)**
>
> Thank you very much for taking the time to review our paper.
> Thank you also for summarizing the paper and clearly outlining its strengths.
> We have addressed the questions and concerns you raised as thoroughly as possible, and we hope our responses adequately resolve them.
> In addition, we will carefully correct the minor issues you pointed out during the revision.
> We would greatly appreciate your participation in the discussion phase as well, and we welcome any further comments you may have.
>
>
> >Lack of clarity in the theoretical part. The theoretical statements are all expressed in dense natural language, making it hard to isolate the main claims and their assumptions. There is a lack of formal structure (e.g., lemmas or theorems) that would help organize the theoretical contributions.
>
> Thank you for the advice.
> We appreciate your suggestion regarding the need for a clearer formal structure.
> In the revised manuscript, we will carefully restructure the theoretical part to improve clarity and present the theoretical contributions in a more formal and transparent manner.
>
> >In particular, $\alpha_t$ a key quantity in the analysis, is never formally defined. It is unclear whether it corresponds to the classical convolution kernel of the SSM or to another object. It is introduced (L172) after a citation to Dambre et al. (2012), but the precise result being referenced from that work is not specified.
> The authors also state that “since the MF  is statistically equivalent to the norm of ” (L176) without citing where this result originates from.
>
> Thank you for this important comment. Although the quantity $\alpha_t$ was first introduced in Jaeger, (2001a).
> the explicit representation of the state $x_t$ as a polynomial expansion of delayed input signals was in fact presented by Kubota et al. (2021).
> We acknowledge that our manuscript lacked the necessary references for fully understanding these results.
> We will revise the manuscript to include the proper citations.
> These studies demonstrate that the information processing of any deterministically evolving dynamical system can be statistically evaluated through a polynomial expansion of the delayed input series.
> When focusing on the terms in a linear combination of delayed inputs (e.g., $u_{t}$, $u_{t-10}$, and $u_{t-37}$),
> their coefficients corresponding to each delayed input are the MF,
> which coincides with the definition of MF presented in this manuscript.
> When including nonlinear terms of the input (e.g., $u_{t-3}^2$, $u_t u_{t-10}$, and $u_{5} u_{11} u_{t-37}^5$),
> the coefficients of the terms represented by these inputs are referred to as information processing capacity (IPC).
> Here, we adopted this concept and, following prior work, expanded $y$ as a polynomial of the delayed inputs and use this notation.
>
>
> >S4D is equivalent to S4. The assumptions underlying this claim are not discussed. It seems to rely on the uniform i.i.d. input assumption, which should be made explicit and justified.
>
> Thank you for the comment. We will ensure that the revised manuscript includes a justification for introducing the i.i.d. input assumption, as you suggested.
> Altough we described the input as uniform distribution here to maintain consistency with the numerical experiments presented in the main text,
> in fact, the analytical results derived in the manuscript hold for any i.i.d. input distribution.
> MF is a statistical measure for evaluating information processing and, of course, cannot exactly represent the memory of correlated inputs other than i.i.d. inputs.
> However, we believe that this simplification provides an important theoretical justification for why there is little performance difference between S4 and S4D.
> To justify that this assumption, which may limit the applicability of our results, is nonetheless reasonable,
> we performed experiments to confirm the predictions indicated by our theoretical analysis.
>
> >Moreover, according to the empirical results reported in the paper (and in previous literature on S4 and its diagonal variants), S4 outperforms S4D. It would therefore be important to clarify under which conditions this theoretical equivalence is supposed to hold.
>
> Thank you for the comment. The equivalence we discuss is derived from the viewpoint of MF as a statistical measure of information processing under i.i.d. inputs.
> Therefore, this equivalence remains purely theoretical.
> The numerical equivalence, on the other hand, must be confirmed empirically.
> As in prior work, our experiments also show that the results of S4D are slightly worse than those of S4,
> suggesting that the theoretical equivalence may not always hold in empirical sense.
> We will explicitly include this point in the conclusion.

---

> > ### Author Response · Authors · 2025-11-18
> > **Response to Reviewer kJk2 (2/3)**
> >
> > >Initialization for successful learning. The derived gradient formula (Eq. 12) is not clearly connected to the final claim that one should prioritize MF profiles with longer delays (even if their amplitudes are small).
> >
> > Thank you for the comment.
> > As explained in the paragraph starting at line 259 of the manuscript,
> > Eq. 12 immediately demonstrates that, at initialization, if the model does not possess the required memory,
> > it is theoretically impossible to extend the memory length explicitly through learning.
> > This paragraph illustrates this situation with a concrete example.
> > Specifically, we first consider a task that requires information only from the very distant past, and assume that the model is unable to retain memory in this domain at initialization.
> > Analytically, due to the action of the MF, the model can never acquire the information necessary to solve this task.
> > As a result, it consistently outputs zero for the component corresponding to the label information $V^{\top}(V V^{\top})^{-1}V \alpha$, preventing any learning progress.
> > From this, as stated at line 268, it follows that solving an arbitrary task would require the model to possess infinite memory length at initialization.
> > Moreover, since the sum of the MF is bounded, memory length must be prioritized over precision.
> > To facilitate understanding, we recommend referring to the definition of $y$ on line 212, as well as Appendices D.2 and D.3, which provide helpful clarification.
> >
> > >If this is a central claim, it could be empirically tested by comparing systems along the Pareto front of MF profiles (accuracy vs. long-delay memory) to verify whether emphasizing long-delay memory indeed improves learning.
> >
> > Thank you for this important comment.
> > As a premise, there is currently no established method for designing a desired MF and the constraints on the shapes of the MF remain unclear.
> > Under the linear SSM formulation, any attempt to extend the MF has, at least in our numerical experiments, inherently led to an increase in its precision.
> > These indicate that a long and low-precision MF is currently a theoretical requirement,
> > but could actually exist.
> > Therefore, the claim you mentioned also remains theoretical prediction.
> >
> >
> > >Fragile link between theory and experiments. The main empirical claim—that freezing the recurrence improves performance—is only loosely justified by the theory.
> > The explanation of an “unintended extension of the MF” (L302) would be stronger if formalized or tested empirically.
> >
> >
> > Thank you for your constructive advice.
> > The necessity of conducting experiments in the RC setting arises from two opposing predictions about learning full parameters,
> > which are derived from our theoretical results.
> > One is the occurrence of unintended extension of the memory function, and the other is the potential decrease of memory in accordance with the capacity required by the task.
> > This issue is fundamental to the optimization of memory capacity in SSMs, and we consider it essential to empirically verify it in order to achieve better model optimization.
> >
> > Although the concept of unintended extension is important, its explicit definition was omitted in the main text due to space constraints.
> > As suggested, we will explicitly and clearly define unintended extension, and will include this definition in the main text or the appendix in the revised manuscript.
> >
> > Definition:
> > Assume that the input information $u_{t-\tau_1}$ is necessary in order to predict the label information $y_t$ in a task,
> > and that, for $u_{t-\tau_1}$, the MF satisfies $M[u_{t-\tau_1}] > 0$.
> > Then, consider the MF for a more distant past input $u_{t-\tau_2}$ with $\tau_2 > \tau_1$, i.e., $M[u_{t-\tau_2}] \geq 0$.
> > Unintended extension refers to the situation where, during training, the model tries to improve $M[u_{t-\tau_1}]$,
> > and as a side effect, $M[u_{t-\tau_2}]$ also increases.
> >
> > Here, we focus on a particularly important case where, $M[u_{t-\tau_2}] = 0$ before training.
> > We empirically confirm in Fig.~3 that, in this case, unintended extension indeed occurs.
> > The paragraph starting at line 421 of the main text is dedicated to explaining both the occurrence of this phenomenon and its effects.

---

> > > ### Author Response · Authors · 2025-11-18
> > > **Response to Reviewer kJk2 (3/3)**
> > >
> > > >If the underlying mechanism is instead regularization or overfitting mitigation, the authors should compare against standard techniques.
> > >
> > > Thank you for the comment. We will address this concern together with our response to Q.2.
> > > We have explained that, from a theoretical standpoint, a longer MF offers only advantages.
> > > Therefore, suppressing the extension of the MF is not required for the arguments developed in this study.
> > > Moreover, because the central claim of this work is that the MF cannot be explicitly extended, we consider MF extension to be unrelated to regularization or overfitting.
> > > For this reason, comparing our mechanism with the methods would not be meaningful from the perspective of understanding the underlying mechanism.
> > >
> > > However, such a comparison could be meaningful for performance improvement.
> > > Although we do not conduct this analysis here, we believe that more careful tuning remains possible under the RC setting.
> > > In this study, we adopted the hyperparameters used in prior work.
> > > We keep the configuration consistent across tasks and no model-specific tuning is performed, changing only the random seed.
> > > Since weight decay and dropout were used in the prior work, we incorporate both in our experiments as well.
> > > We consider the hyperparameter settings presented in the prior work to be close to optimal, and therefore believe that further performance improvements are extremely difficult under the conventional full-parameter training framework.
> > > In contrast, we consider that the performance under the RC setting may still be improved.
> > >
> > >
> > > >Reservoir computing motivation. The appeal to reservoir computing (RC) as motivation for not training the recurrence is weak. In RC, the motivation is computational efficiency (avoiding backpropagation through time), not improved accuracy. Here, since gradients are still computed through the recurrence, this rationale is only partly convincing.
> > >
> > > Thank you for the comment.
> > > As mentioned in our previous comment,
> > > we refer to the setting as RC to simply represent the setting of fixed recurrent weights.
> > > This RC setting is grounded in our theoretical considerations and allows us to verify our predictions.
> > > We did not employ the setting for the purpose of reducing energy consumption.
> > >
> > > >A more convincing motivation would be to analyze how to identify and remedy the pathologies that arise when training the recurrence.
> > >
> > > Thank you for the comment. This is exactly true.
> > > In fact, the potential risk of learning eigenvalues is suggested on line 469 by the possibility of their absolute values approaching $1$.
> > > By default, the S4 model has a structure in which the absolute values of all eigenvalues are very close to $1$.
> > > In an RNN, an eigenvalue over $1$ implies divergence. From this perspective, fixing the eigenvalues helps prevent divergence and suppresses unexpected behavior during parameter updates.
> > > An example of absolute eigenvalues before and after training is shown in Fig. 8 in Appendix E.3.
> > >
> > >
> > > >Answer for Q.1. Scope of gradient analysis
> > >
> > > From the perspective of the MF, it is theoretically shown that the input weights can be chosen arbitrarily (line 177).
> > > This implies that, unlike the output weights, the input weights do not require learning and can remain fixed.
> > > Indeed, we also incorporated this setting when RC setting is applied in the numearical experiments.
> > > We consider that no further investigations regarding the input weights are necessary,
> > > but thank you for the careful read.

---

### Official Review · Reviewer_JNnW · 2025-10-31

**Soundness:** 3
**Presentation:** 4
**Contribution:** 4
**Rating:** 8
**Confidence:** 3

**Summary:**

This paper presents an analysis of the learning dynamics of (non-selective) state-space models like S4. They analyze the gradient of an SSM to show that the model's "memory function" (MF) multiplies a key term in the gradient calculation. If a model's MF is zero for a long time delay at initialization, the corresponding component of the gradient will also be zero, making it impossible to learn the long-range dependency. They argue that these initial weights are then of critical importance; training them can "destroy" the memory (but also possibly expand it?). They propose a "reservoir computing" setting (RC) instead, where the recurrent weights/eigenvalues are kept fixed throughout training. They show this actually works better than fully training the model, converging faster, reaching lower loss, and having better validation accuracy.

**Strengths:**

- Strong theoretical justification for why the initialization of SSMs is so critical: the finding that the MF multiplies the teacher information in the gradient seems important and relatively intuitive and moves beyond the usual vanishing/exploding arguments
- The RC setting is non-obvious and compelling given the analyses
- The empirical support is quite strong for the particular problem of modeling long sequences (in synthetic settings); that it fails with random initializations provides a natural additional baseline. Generally nice experiment design.
- The proof of equivalence between S4 and S4D seems like an important contribution

**Weaknesses:**

- The analyses are for linear SSMs like S4/S4D and cannot be easily extended to more practical models like Mamba (but I suspect the contribution is relatively transferrable)
- Performance degradation on the Listops task that seems to be unexplained

**Questions:**

The idea that the initialization is critical for ensuring the gradient as non-zero signal for long-range dependencies seems related to other empirical findings such as [0], which initializes Mamba's $A$ matrix for "full history" to improve recall and copying performance. Do the authors think that their theory could help explain why this works, even if the particular math doesn't apply to the Mamba setting?

Do you have any other speculation on if this might apply to larger-scale selective SSMs as well? (To be clear, I don't think larger-scale experiments are necessary to warrant acceptance; the paper seems well-scoped.)

Did you attempt to find any more optimal initializations, beyond the additional random/linear schemes?

[0] Trockman, A., Harutyunyan, H., Kolter, J. Z., Kumar, S., & Bhojanapalli, S. (2024). Mimetic initialization helps state space models learn to recall. arXiv preprint arXiv:2410.11135.

---

> ### Author Response · Authors · 2025-11-18
> **Response to Reviewer JNnW**
>
> We sincerely appreciate the reviewer for taking the time to read the manuscript carefully. Thank you for the thoughtful understanding and positive recognition of the strengths of our work.
> We have carefully addressed the concern and question you raised, and we hope our explanations clarify the points. We would be grateful for your continued insights during the discussion phase and welcome any further feedback you might have.
>
> >The analyses are for linear SSMs like S4/S4D and cannot be easily extended to more practical models like Mamba
>
> In fact, our theoretical results based on MF could potentially be directly applied to nonlinear SSMs such as Mamba.
> In Mamba, the SSM can be interpreted as a nonlinear RNN.
> Because the MF can be assessed similarly, the insights on initialization and the importance on memory length may remain valid.
> However, analyzing nonlinear RNNs not only requires focusing on MF but also the memory arising from the nonlinear information processing.
> Unlike in linear RNNs, it remains unclear which factors are critical for achieving high performance in nonlinear memory.
> Identifying what aspects of memory are essential for nonlinear RNNs is an ongoing research topic.
> For this reason, we restrict our analysis to linear RNNs in the present study.
> The measure for investigating nonlinear information processing already exist and are referred to as
> information processing capacity (IPC) that is an nonlinear extension of MF.
> It is potentially possible to measure the IPC of the nonlinear SSM within Mamba.
>
>
> >Performance degradation on the Listops task that seems to be unexplained
>
> Thank you for pointing out this.
> We will add a more thorough discussion of these results in the revised manuscript.
> The necessity of conducting experiments in the RC setting arises from two opposing predictions about learning full parameters,
> which are derived from our theoretical results.
> One is the occurrence of unintended extension of MF, and the other is the potential decrease of memory in accordance with the capacity required by the task.
> As noted in the manuscript, a slight improvement in memory is observed through the learning of eigenvalues.
> This implies that the performance on Listops may arise from this slight memory improvement.
>
>
> >Questions:
>
> >The idea that the initialization is critical for ensuring the gradient as non-zero signal for long-range dependencies seems related to other empirical findings such as [0],
> which initializes Mamba's matrix for "full history" to improve recall and copying performance.
> Do the authors think that their theory could help explain why this works, even if the particular math doesn't apply to the Mamba setting?
>
> Thank you for providing references that are also useful for our study.
> We are currently reading the paper carefully to deepen our understanding.
> Once we have sufficiently understood them and can respond appropriately, we will contact you again.
>
>
> >Do you have any other speculation on if this might apply to larger-scale selective SSMs as well?
> (To be clear, I don't think larger-scale experiments are necessary to warrant acceptance; the paper seems well-scoped.)
>
> Thank you for the question.
> In the context of our study, selective SSMs can be collectively evaluated as nonlinear RNNs under the IPC metric.
> These models may share specific nonlinear information processing mechanisms that contribute to their performance improvements in addition to the linear information processing that S4 already has.
> From the perspective of IPC and MF, the upper bound of IPC is determined by the number of nodes.
> Therefore, if S4 and Mamba have the same number of nodes in their SSMs, their theoretical maximum IPC is also the same.
> However, in practice, the total effective IPC can differ from this theoretical upper bound. This total can be regarded as the number of effective nodes. Making the RNN nonlinear may increase the number of effective nodes. Additionally, nonlinear operations may also enhance the diversity of linear information processing. These effects together suggest that a more desirable information processing may be realized in practice.
>
>
> >Did you attempt to find any more optimal initializations, beyond the additional random/linear schemes?
>
> We attempted to derive an analytical solution that achieves the optimal MF structure,
> but this has not yet been achieved.
> We are currently exploring the possible shapes with either long or short MFs through numerical experiments.
> As mentioned at the end of the manuscript, although the improvement in memory through learning is small,
> it is nonetheless possible, and this motivates the proposal of better initializations.
> Currently, no eigenvalues that clearly realize a better MF have been identified,
> and thus they could not be incorporated in this experiment.
> However, if such eigenvalues exist, adopting them as new initializations and fixing them during training would potentially combine the advantages of long MF with those of the RC setting.

---

### Author Response · Authors · 2025-12-03
**Comments to the Area Chair**

Dear Area Chair,

We appreciate your handling of the review process.
We also thank the reviewers for their constructive feedback and for recognizing the importance of our study on learning mechanisms in state space models (SSMs).
The reviewers highlighted several strengths of our study, including the novelty of our perspective on SSMs based on the memory function (MF), the usefulness of stabilizing learning by fixing the eigenvalues of the recurrent weights, and the importance of our proposed mechanism, which clarifies why initialization is critical for a successful learning.
All reviewer comments have been fully addressed.
The revisions made in the text are highlighted in red.
The main revisions are summarized below.

---

### 1. Clarifying differences from prior work (Reviewer WKGg)

To address the concern regarding unclear distinctions from previous work, we revised the **Introduction** to explicitly describe how our problem formulation differs. We also updated the **Discussion and Conclusion** to highlight the contributions of our study. A detailed methodological and technical comparison is now included in the **Appendices F1 and F2**.
Our clarifications focus on the following two differences.

**(a) Equivalence of SSMs via eigenvalue structure.**
Prior analyses did not establish theoretical equivalence between S4 and S4D from the perspective of memory structure. We show that when recurrent matrices share identical eigenvalues and the system is full rank, the choice of eigenvectors does not affect the information processing in terms of MF. Earlier studies imposed additional constraints on the recurrent matrix or its eigenvalues, which limited the generality of their conclusions.

**(b) Difference from standard vanishing/exploding-gradient analyses.**
Prior work emphasizes that proper initialization helps avoid gradient vanishing/explosion but does not demonstrate that such initialization is *necessary* for successful learning.
In contrast, our work asks whether gradient-based optimization can recover an SSM that begins with an insufficient memory structure.
If a optimization method had a mechanism to restore the loss of information due to the insufficient memory structure during training, initialization would not be critical for a successful learning.
Our analysis shows that no such mechanism exists for SSMs; therefore, the necessary memory structure should already be present at initialization.

---

### 2. Improving formal structure (Reviewer kJk2)

Responding to the comment that the theoretical presentation lacked formal structure, we added a more formal exposition of the results in the **Appendix D4** and clearly referenced these additions in the main text. Other suggestions requiring integration into the main paper were also incorporated.

---

### 3. Extensions to modern SSMs (Reviewers JNnW and NU5d)

Following the reviewers’ recommendations, we modified the **Appendix F4** section explaining how our analysis can be extended to modern architectures, including Mamba, Mamba2, and Gated DeltaNet.

---

We believe that these revisions clarify the contributions and adequately address all reviewer concerns. We appreciate your time and consideration.

---

### Meta-Review · Area_Chair_VMxJ · 2026-01-07

**Summary:**

The paper analyze the impact of initialization at the convergence and training of SSMs. Their analysis lead to a new optimization strategy that is relevant for SSMs. The authors discuss the impact on Mamba and other models. While interesting, the problem studied is very much related to vanishing gradients and other analysis that was done in the literature and therefore the novelty of the work is quite limited in scope. Therefore, the paper cannot be accepted as the rebuttal did not properly address these claims that were raised by the reviewers and especially WKGg

**Reviewer Concerns:**

The reviewers raised concerns with respect to the scope of the theory proposed, the relation to selective SSMs like Mamba and the relation to prior work. Specifically, the most concerning concern was related to the fact that the results presented may be simply interpreted as a vanishing gradients problem, which has been studied before vastly. The reviewers and especially WKGg pointed to several relevant works and I did not see a proper response to these works.

**Reviewer Scores:**

Some of the claims were properly answered but the most pressing concern was not and reviewer that raised it mentioned explicitly the dissatisfaction from the answer with which I fully agree. Given this important concern the paper cannot be accepted and need to be significantly revised to explain the relation to prior work and re-emphasize its scope and contribution and difference from prior vanishing gradient analysis (is there a major difference?)

---

### Decision · Program_Chairs · 2026-01-26

Reject